# A thalamic-hippocampal CA1 signal for contextual fear memory suppression, extinction, and discrimination

Heather C. Ratigan[1,2,3], Seetha Krishnan [1,3], Shai Smith [1,4] & Mark E. J. Sheffield [1,2,3,4] ✉

The adaptive regulation of fear memories is a crucial neural function that prevents inappropriate fear expression. Fear memories can be acquired through contextual fear conditioning (CFC) which relies on the hippocampus. The thalamic nucleus reuniens (NR) is necessary to extinguish contextual fear and innervates hippocampal CA1. However, the role of the NR-CA1 pathway in contextual fear is unknown. We developed a head-restrained virtual reality CFC paradigm, and demonstrate that mice can acquire and extinguish context-dependent fear responses. We found that inhibiting the NR-CA1 pathway following CFC lengthens the duration of fearful freezing epochs, increases fear generalization, and delays fear extinction. Using in vivo imaging, we recorded NR-axons innervating CA1 and found that NR-axons become tuned to fearful freezing following CFC. We conclude that the NR-CA1 pathway actively suppresses fear by disrupting contextual fear memory retrieval in CA1 during fearful freezing behavior, a process that also reduces fear generalization and accelerates extinction.

Flexibly encoding and retrieving memories of fearful events is a critically conserved survival behavior, as a single failure can be deadly. However, failing to suppress inappropriate fear responses can also have devastating consequences, manifesting as negative affective states in generalized anxiety disorder and post-traumatic stress disorder[1,2]. One way in which fear memories can be studied in the laboratory is through contextual fear conditioning (CFC), in which a spatial context, the conditioned stimulus (CS), is repeatedly paired with a noxious unconditioned stimulus (US), generally a mild shock[3–6]. Freezing is a species-specific fear response, and a quantifiable readout of contextual fear memory retrieval (CFMR) of the learned association[7,8]. With continued exposure to the CS in the absence of the US, freezing generally decreases and exploratory behavior increases - a process termed fear extinction. Fear extinction occurs as animals learn that the context no longer predicts shocks[9–12].

Contextual fear in both mice and humans relies on coordinated brain regions including the medial prefrontal cortex (mPFC), thalamus, amygdala, and hippocampus[13]. The contextual component of these memories relies on the hippocampus, which retrieves and updates contextual fear memories[6,14–19]. Following CFC, experimental inhibition of a subset of hippocampal neurons tagged using immediate early genes active during CFC is sufficient to suppress CFMR[20–22]. This suggests that natural suppression of ongoing CFMR must involve a circuit that can modulate hippocampal activity.

One potential source of this modulation is the ventral midline thalamic subregion, nucleus reuniens (NR). Sometimes termed 'limbic thalamus' for its diverse set of inputs from limbic-related regions in the brainstem, hypothalamus, amygdala, basal forebrain, mPFC, entorhinal cortex (EC), and hippocampal subregion CA1, NR sits at the nexus of emotional regulation and serves as a major communication hub

[1]Department of Neurobiology, University of Chicago, Chicago, IL 60615, USA. [2]Doctoral Program in Neurobiology, University of Chicago, Chicago, IL 60615, USA. [3]Neuroscience Institute, University of Chicago, Chicago, IL 60615, USA. [4]Undergraduate Program in Neuroscience, University of Chicago, Chicago, IL 60615, USA. ✉e-mail: sheffield@uchicago.edu

among these limbic-activated areas[23–28]. While mPFC does not have a direct excitatory projection to CA1, it does send a strong excitatory projection to NR[23,24].

The mPFC-NR projection and NR itself are necessary for both fear extinction and for preventing fear generalization to a neutral context, a process in which animals associate a non-shocked context with fear[29–35]. NR stimulation reduces contextual fear-induced immediate early gene expression in both mPFC and CA1[35,36]. While the roles of the mPFC-NR pathway and NR itself have been explored during CFMR, the role of the NR-CA1 pathway is unknown. We hypothesize that NR transmits a signal to CA1 to suppress ongoing CFMR, thereby reducing fear responses (freezing) and promoting exploratory behavior (movement).

To test our hypothesis, we used a chemogenetic approach to directly inhibit the NR-CA1 pathway, and 2-photon calcium imaging in head-restrained male mice to record NR-axons in CA1, before and after CFC. While CFC induction in head-restrained mice in virtual reality (VR) has been attempted, none to our knowledge have replicated the characteristic 'freezing' behavior of freely-moving mice in real-world CFC[37,38]. We therefore developed a new VR-based CFC paradigm (VR-CFC), using a conductive fabric to deliver mild tail shocks that induces context-dependent freezing. By combining VR-CFC, targeted chemogenetic NR-CA1 inhibition, and 2-photon NR-axonal calcium imaging, we were able to determine the role of the NR-CA1 pathway in CFMR, generalization, and extinction.

## Results

### Contextual fear conditioning and extinction in virtual contexts

To ensure mice were comfortable with the VR setup before shocks were delivered, we trained water-restricted mice to run in a VR context for water rewards until they reached ~4 traversals of the context per minute, as previously discussed[39,40]. To avoid confounds from the water reward during CFC, these trained mice were then introduced to two novel VR contexts without a water reward. At this stage, the custom-designed conductive tailcoat was fitted to their tails (Fig. 1a, Methods: Behavior). Mice then spent ~5 minutes in each novel VR context, which allowed them to habituate to running with the tailcoat (Fig. 1b; Methods: Behavior).

Mice in all experimental conditions that continued to meet the criterion for movement (i.e., >4 traversals per minute) were advanced to the next stage the following day (46/79 mice), where they were re-exposed to both novel contexts for ~5 minutes each. During this period, mice demonstrated low levels of spontaneous freezing behavior (Supplementary Fig. 1f). Mice were then administered 6 mild 0.6 mA tail shocks through the tailcoat for a duration of 1 s each, 20-26 s apart, (Fig. 1b: day 0). These shocks were delivered at pseudorandom locations throughout one of the contexts ('shocked'), but not the other ('control'; Fig. 1h). Mice responded to each tail shock with an abrupt stereotyped increase in running speed, a behavioral validation of successful shock delivery (Fig. 1c). To test for CFMR and subsequent fear memory extinction, mice were then re-exposed for ~5 minutes each, in a pseudorandom order, to both the shocked and control contexts while wearing the tailcoat for the following three retrieval days (Fig. 1b: day 1–3). Elevated freezing behavior on post-shock days was interpreted as an expression of CFMR (Methods: Behavioral Parameters - Freezing).

In the shocked context on retrieval day 1, mice ($N = 20$) froze 32.7 ± 4.7% (95% CI) more than the pre-shocks baseline day, a statistically significant increase, while mice froze in the control context on average 8.7 ± 4.7% more than the baseline day, a non-significant increase (Fig. 1e). Comparing across contexts, mice increased their freezing significantly more in the shocked context compared to the control context, showing context specificity in CFMR (Supplementary Fig. 1d). This remained true on retrieval day 2, as mice continued to freeze at significantly elevated levels above-baseline in the shocked

context and compared to the control context (Fig. 1e and Supplementary Fig. 1d). By the third day of retrieval, mice froze at comparable levels between contexts (Supplementary Fig. 1d), although remained slightly elevated above-baseline in the shocked context (Fig. 1e).

While freezing levels differed between contexts and days of retrieval, freezing position was distributed evenly across all track locations in both contexts on all retrieval days. This shows that mice associated fear with the entire context, and not specific locations along the track or near specific VR objects (Fig. 1i). As an additional control, a separate group of mice went through the same process but were never shocked in either context. These mice had similar baseline freezing levels to the VR-CFC mice, but did not have any significant differences in freezing levels on subsequent days compared to baseline or compared across contexts (Fig. 1f; Supplementary Fig. 1f). The low level of spontaneous freezing epochs in both the non-shocked control group and the pre-shocked contexts (i.e., before the delivery of any shocks) could potentially be caused by the lack of water reinforcement, the presence of the tailcoat itself, or a temporary disinterest in running, and provides a useful within-mouse comparison to post-shock fear-evoked freezing.

To further quantify freezing behavior, we measured the duration of each individual freezing event (freezing epoch) and found that freezing epochs were longer in the shocked context at an average of 5.9 s compared to 4.6 s in the control context per freezing epoch on retrieval day 1 (Fig. 1g). Freezing epochs also remained significantly longer on day 2 in the shocked compared to the control context, however, they became similar by day 3 (Supplementary Fig. 2b–d), corresponding with trends in the total time spent freezing. We additionally examined if VR-CFC impacted non-freezing running behavior, and showed that average non-freezing velocity in either context compared to baseline or between contexts was not statistically impacted by VR-CFC, indicating that VR-CFC selectively impacts freezing behavior (Fig. 1j). Our results show that VR-CFC produces robust CFMR that can be measured via context-specific increases in freezing, which is extinguished following ~3 days of re-exposure to the shocked context in the absence of additional shocks.

### Inhibition of the NR-CA1 pathway during CFMR

To test the involvement of NR-CA1-projecting neurons in CFMR, we developed a designer receptor exclusively activated by designer drugs (DREADD) based inhibition paradigm[41–43] (Fig. 2a). We injected a Cre-expressing virus bilaterally in NR, and a retrograde Cre-dependent virus carrying the inhibitory G(i)-coupled DREADD receptor, hM4Di-DREADD, bilaterally in the SLM of dorsal CA1 where hippocampal-projecting NR-axons terminate. This led to the expression of DREADDs selectively in NR neurons projecting to CA1 and can be seen in NR axons in SLM of CA1 (Fig. 2b)[44]. This enabled us to intraperitoneally (IP) inject the hM4Di agonist, deschloroclozapine dihydrochloride (DCZ), before the first post-shock re-exposure to the contexts on retrieval day 1, therefore selectively inhibiting a subset of NR-CA1-projecting neurons during CFMR. DCZ was selected over CNO for its comparatively higher DREADD-selective binding at lower doses, reduced off-target effects due to reduced conversion into clozapine, and rapid onset kinetics.[43]

To ensure our injection paradigm and administration of DCZ did not alter context-dependent fear behavior, we had two DREADD control groups. One group ($N = 4$) expressed mCherry in place of hM4Di, and received DCZ on retrieval day 1. A separate group ($N = 4$) expressed the hM4Di receptor, and received saline instead of DCZ on retrieval day 1 (Supplementary Fig. 1g). In both control groups, freezing behavior was similar to the experimental mice shown in Fig. 1e, and the groups were thus combined with other NR-CA1 uninhibited mice and termed the NR-CA1 intact group for analysis. We then examined the behavioral impact of inhibiting the NR-CA1 pathway on day 1, and on subsequent retrieval days 2 and 3 (Fig. 2e).

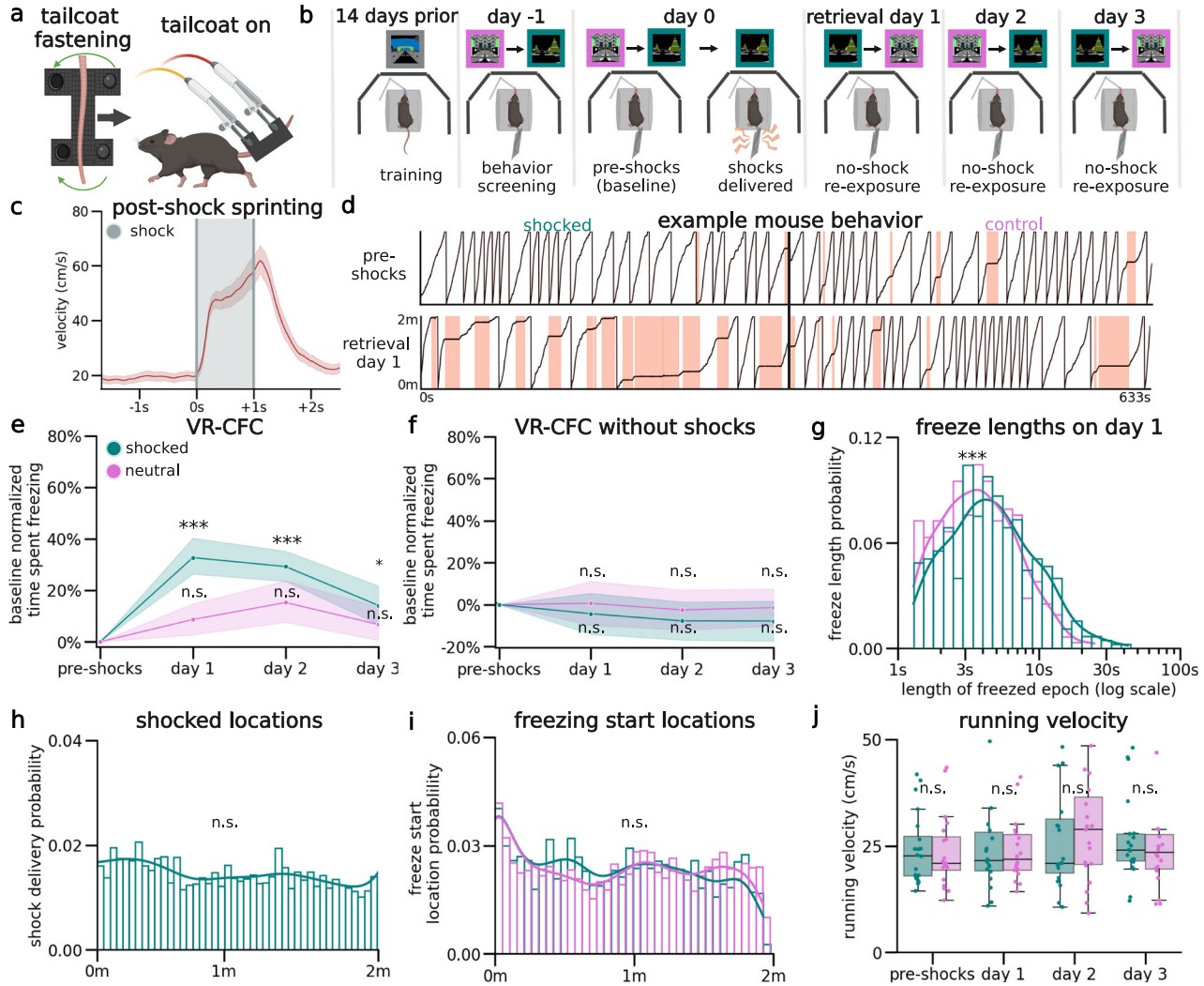

**Fig. 1 | Virtual reality contextual fear conditioning induces robust fearful freezing responses. a** Schematic of tailcoat. Left: 1.8 g tailcoat is comprised of conductive cloth secured around the tail. Right: the tailcoat is suspended using alligator clips forming a weight-supportive 'hammock' structure connected to an electric shock generator. **b** After training, head-restrained mice received mild tail shocks in one unrewarded context (shocked), and not the other (control). They were re-exposed to both for 3 retrieval days. **c** Animal velocity aligned to shock initiation and plotted across shocks shows post-shock sprinting behavior (red line = mean, red shading = 95% CI, gray shading = shock duration). **d** Example mouse position pre-shocks (Top), and on day 1 (Bottom), in the shocked (left) and control (right) contexts. Red shading indicates freezing. **e** Per mouse, we tested the difference in %freezing from baseline within contexts (*N* = 20 mice; CI = 95% shaded area). Shocked mice froze significantly more in the shocked context post-shocks on retrieval days 1–3 (Two-sided Estimated Marginal Mean on ANOVA (EMM), *P* = day 1:

1.16e-8, day 2: 8.65e-8, day 3: 0.04), but not the control context (EMM, *P* = day 1: 0.76, day 2: 0.07, day 3: 1.). **f** Unshocked mice on day 0 froze continuously at baseline levels, (*N* = 7, EMM, Shocked context: *P* = 1). **g**, Kernel density estimates and density histogram of the length of freezing epochs show they were shorter in the control than the shocked context on day 1 (Mann–Whitney *U*, P = 1.20e-3). **h** Density histogram of shock locations show shocks were administered evenly across the track (Kolmogorov–Smirnov, *P* = 0.91). **i** Density histogram of freeze start locations across the virtual track indicate that mice freeze indistinguishably at all locations (Kolmogorov–Smirnov, *P* = 0.89), in both contexts (Mann–Whitney *U*, *P* = 0.24). **j** Mean running velocity in each mouse when not freezing (dots, *N* = 20). Boxplot indicates median, 25–75th interquartile range, whiskers include all data points No differences were observed (Student's T, P = pre-shocks: 0.60, day 1: 0.60, day 2: 0.30, day 3: 0.57). Created with BioRender.com.

We found that in the shocked context on retrieval day 1, NR-CA1 inhibited mice (*N* = 9) spent 58.5 ± 10.4% more time freezing on day 1 under the influence of DCZ in the shocked context when compared to their pre-shock baseline freezing levels, with every mouse increasing their freezing level in amounts ranging from a minimum of 41.3% to a maximum of 84.6% (Fig. 2d). Freezing in the control context also significantly increased from baseline to 31.0 ± 15.7%. under the influence of DCZ (Fig. 2e). Freezing levels on day 2 remained significantly elevated in the shocked context on day 2 in the absence of DCZ at 48.3 ± 9.0%, while freezing in the control context fell to baseline levels at 14.8 ± 11.1% (Fig. 2e). Freezing levels on day 3 remained slightly elevated in the shocked context, at 23.4 ± 9.1% while remaining not significantly different from baseline in the control context at 8.44 ± 8.5%.

Overall, NR-CA1 inhibition induced large increases in freezing levels in both contexts, which persisted beyond the DCZ delivery day in the shocked context.

## NR-CA1 inhibition elevates freezing, reduces context discrimination, and delays extinction

To further test the impact of NR-CA1 inhibition, we directly compared freezing behavior in NR-CA1 intact mice to NR-CA1 inhibited mice (Fig. 3). We found that NR-CA1 inhibited mice under DCZ froze ~78% more in the shocked context (Fig. 3a) and ~256% more in the control context (Fig. 3b) compared to NR-CA1 intact mice on day 1. This elevated freezing response remained true for the shocked context on day 2, with a ~65% increase compared to intact mice,

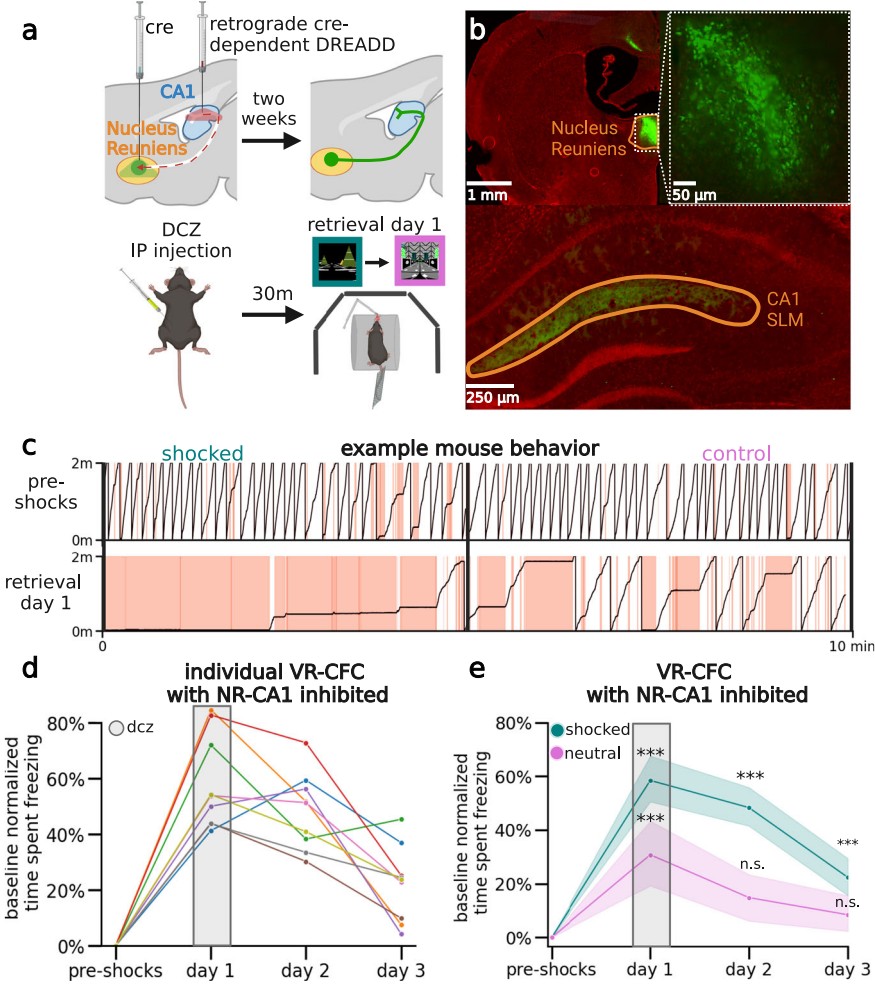

**Fig. 2 | Nucleus reuniens-CA1 pathway inhibition one day following CFC. a** The NR-CA1 pathway was inhibited by first injecting retrograde CRE-dependent DREADDs in SLM of CA1 and CRE in NR and then injecting DCZ systemically 2 weeks later. Bottom: ~30 minutes before context re-exposure on retrieval day 1 (1 day after CFC), mice received 0.1 mg/kg of the HM4d agonist DCZ. **b** Example confocal imaging of HM4di+Cre expression (total $N = 9$) in NR and SLM of CA1 in a coronal section from an example mouse. HM4di expression is labeled in green, DAPI in red. Top left: Section with Nucleus Reuniens (orange) showing DREADD expression selectivity to NR (green). Top right: NR section zoomed in to highlight expression in individual neurons. Bottom: CA1 hippocampal section showing DREADD expression in NR axons in CA1 restricted to the SLM. **c** Example track position from a mouse on day 0 pre-shocks (Top), and on retrieval day 1 with NR-CA1 inhibited (Bottom), in the shocked (left) and control (right) contexts. Red shading indicates freezing epochs. **d** Percent time freezing normalized to baseline for each mouse plotted individually across days in the shocked context. DCZ administration is shown highlighted in gray on retrieval day 1. **e** Teal line shows summary data of 2d showing mean time spent freezing across days with NR-CA1 inhibited on day 1. Pink line shows time spent freezing in the same mice but in the control context ($N = 9$ mice). Shaded areas indicate 95th confidence interval. Freezing levels stay elevated in the shocked context on days 1–3, but are only elevated on day 1 in the control context. (Two-sided estimated marginal mean on ANOVA, shocked $P =$ day 1: 4.15e-10, day 2: 2.34e-9, day 3: 9.40e-3, Control $P =$ day 1: 2.93e-4, day 2: 0.99, day 3: 1). Created with BioRender.com.

even though DCZ was no longer inhibiting the NR-CA1 pathway (Fig. 3a). Thus, inhibiting the NR-CA1 pathway on day 1 reduces the amount of extinction on day 2. However, in the control context on day 2, freezing levels returned to baseline and were not different between the intact and inhibited groups (Fig. 3b). By day 3, freezing levels in both the shocked and control contexts in NR-CA1 inhibited mice were not statistically different from their NR-CA1 intact counterparts (Fig. 3a, b).

We next compared the freeze length distributions between NR-CA1 inhibited and NR-CA1 intact groups to see if an increase in average freeze length was driving the observed increased time spent freezing (Fig. 3c, d). Indeed, this was the case, as average freeze lengths nearly tripled in NR-CA1 inhibited mice, increasing 173% from 5.9 s to 16.2 s in the shocked context on day 1 (Fig. 3c). A similar increase was observed in the control context on day 1, with average freeze lengths increasing 156% from 4.6 s to 11.8 s in NR-CA1 inhibited mice. These findings suggest that inhibiting the NR-CA1 pathway during CFMR increases the

time spent in an ongoing fearful state of freezing by lengthening individual freezing epochs. This suggests that the intact role of the NR-CA1 pathway suppresses CFMR in both appropriate (shocked) and inappropriate (unshocked control) contexts, to reduce fearful freezing.

Given these findings, we asked if mice could still discriminate between the shocked and the control context after NR-CA1 inhibition. To do so, we calculated a discrimination index (DI)[29] which revealed a significant decrease in discrimination between the shocked and control contexts on day 1, going from 37.7 ± 8.1% in NR-CA1 intact mice to 14.2 ± 7.25% with NR-CA1 inhibited (Fig. 3e). No significant difference in the DI was observed on any other day. This result suggests that during inhibition of the NR-CA1 pathway, fear-induced contextual discrimination is reduced (Fig. 3e).

We wanted to ensure that DCZ-induced increases in freezing were not due to a general decrease in movement. To do so, we exposed NR-CA1 inhibited and intact mice to a 'dark' context (devoid of any visual

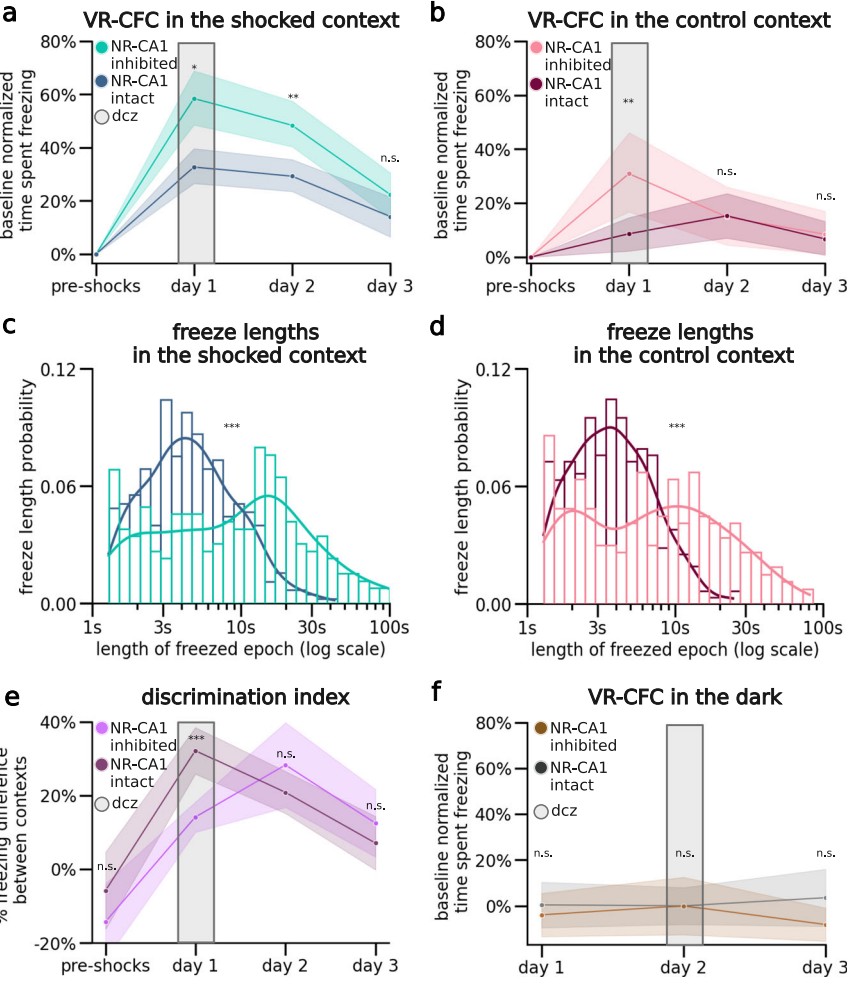

**Fig. 3 | Nucleus reuniens-CA1 pathway inhibition increases freezing, reduces context discrimination, and delays extinction. a** Direct comparison of freezing behavior in NR-CA1 inhibited (same data as Fig. 1e: shocked context; blue) versus NR-CA1 intact (same data as Fig. 2e: shocked context; green) mice (Two-sided Estimated Marginal Mean on ANOVA (EMM), P = day 1: 0.027, day 2: 1.63e-3, day 3: 1). **b** Same as **a**, but for the control context (EMM, P = day 1: 2.18e-3, day 2: 1, day 3: 1). **c** The length of individual freeze epochs in the shocked context with NR inhibition skewed longer compared to NR-intact on day 1 (Mann–Whitney U, P = 2.49e-15). **d** Same as **c**, but for the control context (Mann–Whitney U, P = 5.49e-12). **e** Discrimination index was calculated per day for both the NR-CA1 intact mice and NR-CA1 inhibited mice as (% time spent freezing in shocked context−% time spent

freezing in control context)/total % time spent freezing in both contexts. Center lines indicate mean. Shaded areas indicate 95th confidence interval. NR-CA1 inhibition caused mice to discriminate less between the two contexts (Wilcoxon Rank Sum, P = pre-shocks: 0.67, day 1: 1.20e-3, day 2: 0.89, day 3: 0.98). **f** On retrieval days 1–3, we additionally recorded in a 'dark' context–a dark VR with no visual cues–in NR-CA1 inhibited mice and NR-intact mice for the same length of time as the other context exposures. Center lines indicate mean. Shaded areas indicate 95% confidence interval. Mice froze at consistently low levels in the dark context across days, with no difference in freezing levels between groups (Estimated Marginal Mean on ANOVA, P = day 1: 1, day 2: 1, day 3: 0.10). Created with BioRender.com.

cues) for ~5 minutes after they were exposed to both the shocked and control contexts on retrieval days 1–3. In this dark context, mice quickly recovered their running behavior, with NR-CA1 inhibited mice freezing on average only 4.3 ± 3.8% of the time across all 3 days of retrieval. Both within and across mice controls froze at comparably low levels in the dark context (on average under 5%; Fig. 3f). Therefore, neither DREADD inhibition, nor DCZ itself, impacted the mouse's ability to move, and the increase in freezing behavior is specific to when mice are navigating in VR contexts. Our results thus indicate that the increase in freezing behavior in NR-CA1-inhibited mice is not due to motor impairment of DCZ. Our results overall indicate that the intact NR-CA1 pathway sends a potent fear suppression signal-to-CA1, critical for shortening the length of freezing epochs, limiting fear generalization to other contexts, and speeding up extinction. Inhibiting this fear suppression signal, therefore, induces a higher fear response by increasing the length of freezing epochs, increasing fear generalization, and decreasing extinction.

## NR-CA1 axon activity becomes tuned to freezing behavior following CFC

Excitatory NR projections to the hippocampus are restricted to the stratum lacunosum-moleculare (SLM) of CA1 and subiculum; NR does not project to any other hippocampal subregions or layers[44–46]. Previous work stimulating this projection shows it depolarizes CA1 pyramidal neurons across the dorsal-ventral axis and induces firing in multiple interneuron subtypes with dendritic processes in SLM[47–50]. However, the activity of the NR-CA1 projection in vivo during behavior is unknown. To determine the information transmitted directly from the NR to CA1 during CFMR, we performed in vivo 2-photon Ca$^{2+}$ imaging of NR-axons in SLM.

We injected an axon-targeted virus carrying axon-GCaMP6s into NR (Table 1), followed by a cannula window over CA1 as previously described (Fig. 4a[39,40]). Expression in NR was confirmed via histological evaluation following the completion of experiments (Fig. 4a; left). NR-axons could be observed in the SLM of CA1 both post hoc (Fig. 4a,

middle) and during experiments under 2-photon imaging (Fig. 4a, right). We successfully recorded reliable GCaMP6s expression from 1 highly branching NR-axon per mouse ($N = 10$) in hippocampal CA1 during the VR-CFC paradigm (day 0 and retrieval days 1–3). We limited our analysis to a putative single axon per animal since all identified axonal segments within the field of view (FOV) with above-baseline activity were highly correlated, likely due to single, highly branched axons occupying the majority of our imaging FOV (see Methods: Image Processing and ROI Selection).

We found that NR axons switched their activity from untuned sparse activity (Fig. 4b; Top) pre-shocks, to activity highly selective for freezing epochs post-shocks, even after filtering for axons with detectable pre-shock activity (Fig. 4b; Bottom; same axon shown on both days). Since behavior necessarily changes following successful CFC, which induces more and longer freezing epochs, we needed to avoid potential confounds in comparing axon activity during dissimilar freezing epoch lengths before and after CFC. To do so, we quantified axon activity in three different ways.

First, we examined the mean normalized $\Delta f/f$ of peaks in both contexts during running (Fig. 4c, Control: Green, Shocked: Blue) and freezing (Fig. 4c, Control: Orange, Shocked: Red). Axons were normalized each day to near-maximum activity, controlling for any potential differences in amplitudes across days. This analysis shows that pre-shocks, NR-axonal activity in the to-be shocked context was similar between running and freezing epochs, with a slight preference for running epochs, which was slightly elevated in the control context (Fig. 4c). Conversely, post-shocks in both contexts and on all 3 retrieval days, we found that NR-axons had significantly higher mean peak activity during freezing compared to running (Fig. 4c).

Second, we binned freezing epochs into 1 s intervals based on their total length from 1–2 s to 6–7 s, and compared pre-shock to post-shocks activity within those bins, therefore comparing the NR-axonal activity of similar lengths of freezing both pre and post shocks (Fig. 4c, Supplementary Fig. 3e). Freezing epochs that were longer in length than 7 s were not used for this analysis due to the low quantity of such epochs present pre-shock. Within each binned epoch, we trial-aligned activity to the freezing to running transition point (Fig. 4d, black center line). We then compared average NR-axon activity from all axons during freezing (Fig. 4b, d, peach-shaded regions) to activity during running (unshaded regions; Fig. 4d, 3–4 s long freezing epochs shown; all epochs in Supplementary Fig. 3e). Pre-shocks in either context, NR-axons did not significantly modulate their activity between running and freezing epochs (Fig. 4d). However, post-shocks, we found that NR-axons significantly increased their activity during freezing epochs, compared to reduced activity during running epochs. This was observed during all post-shocks freezing epochs, in both the shocked and control contexts (Fig. 4c).

Third, we characterized the dynamics of NR-axon activity within each freezing epoch on pre-shock day 0 and compared to retrieval day 1. To do so we aligned NR-axons by dividing each freezing or running epoch into 5 even bins, each containing a mean normalized $\Delta f/f$ of NR-axon peaks, then took the within-bin mean across all epochs pre and post-shocks. This enabled us to effectively 'stretch' or 'shrink' all epoch lengths to a uniform standard. Using this method, we found that mean axon activity ramped up rapidly in the beginning of a freezing epoch, plateaued, then fell right before freezing transitioned to running (Fig. 4e). Such temporal dynamics were absent during the freezing epochs pre-shocks (Fig. 4e).

We were additionally able to track the same axon across multiple days in a subset of NR-axons ($N = 4$; example axon shown in Fig. 5a) and independently quantified their activity to observe how individual axon dynamics change throughout CFC, retrieval, and extinction (Fig. 5, analysis of all multi-day tracked axons averaged together). We again found similar levels of baseline activity between the freezing and running contexts pre-shocks (Fig. 5b, far-left). On day 1 post-shocks,

these axons significantly increased their activity during freezing, and decreased their activity during running (Fig. 5b, middle-left). This tuning to freezing behavior remained true on day 2 (Fig. 5b, middle-right), but fell back towards baseline levels on day 3, albeit with a small but significant difference between freezing and running remaining (Fig. 5b far-right). This effect is also reflected in the mean normalized $\Delta f/f$ of peaks, with differences in peak $\Delta f/f$ between freezing and running smaller on day 3 (Fig. 5c). The general shape of activity during a freezing epoch remained 'bow' shaped (Fig. 5d). Overall in the multi-day tracked axonal subset, we found very similar results to our overall NR-axon data (Fig. 4), with the notable decrease of axonal activity on retrieval day 3. This decrease corresponds with behavioral extinction of fear responses (Supplementary Fig. 1h), potentially indicating that individual axons may decrease their activity during freezing as the fear-suppressive effect is no longer needed. These results collectively show that NR-axons projecting to CA1 strongly tune their activity to fearful freezing epochs during CFMR, and this post-CFC activity is context-independent.

## Encoding model predicts NR-CA1 axonal activity, but only following CFC

To further quantify the relationship between behavior and NR-axon activity, and to rule out other potential causes driving activity dynamics other than freezing, such as pupil diameter fluctuations or contributions from other behavioral variables, we developed a quantitative encoding boosted trees decision model to predict axonal activity from behavioral and pupil variables. We trained the model using XGBoost[51] to use behavioral information about freezing epochs, running epochs, velocity, location on the track, and pupil diameter to predict NR-axon activity. We separately trained on 80% of traversals and tested on the remaining 20% of traversals in each mouse, on each day, and in each context (Fig. 6; Supplementary Fig. 4; Methods: Boosted Trees Model). Model prediction most heavily relied on behavioral parameters pertaining to whether the mouse was freezing or running, its velocity, and duration passed or remaining within a freezing or running epoch (Fig. 6h, Supplementary Fig. 4b). Overall, the model predicted NR-axon activity well in both the shocked and control contexts on retrieval days post-shocks (with a context/day-combined 0.43 $r^2$ goodness of fit; Fig. 6g), but predicted axonal activity poorly in both contexts pre-shocks (with a context-combined 0.01 $r^2$; Fig. 6g). In the example mouse shown in Fig. 6c–f, the maximum model accuracy pre-shocks was $r^2$ of 0.06 (Fig. 6c) compared to a much higher $r^2$ of 0.86, 0.88, and 0.78 on retrieval days 1, 2, and 3, respectively (Fig. 6d–f).

Because there was variability in the fluorescence signal recorded from the axons, we checked whether model accuracy was related to the signal-to-noise. Indeed, model accuracy was correlated with axon activity - the greater the change in the normalized fluorescence signal from baseline, the better the model performed (Fig. 6b). The model performed significantly above chance in predicting NR-axon signal in 8/10 mice, on retrieval days 1–3. In 2/10 mice, model prediction was poor on retrieval days, due to lower signal-to-noise ratio (SNR) of the fluorescence signal. However, changes in SNR did not account for the poor model performance pre-shock, as model accuracy was still low in animals with higher axon activity. Although overall activity was higher in post-shock days, pre-shock activity in longitudinally-tracked axons reached similar peak heights as in post-shock days (Supplementary Fig. 3a), and all mice included in analysis had at least 2 peaks reaching a minimum of 0.1 $\Delta f/f$ in the recording session, ensuring that poor model performance was not simply due to a lack of signal to predict. In summary, using an encoding model, we demonstrated that NR-axon activity recorded in hippocampal CA1 can be predicted from freezing behavior, but not before the animal is fear-conditioned, revealing the development of predictable structure in NR-axon activity tuned to CFMR.

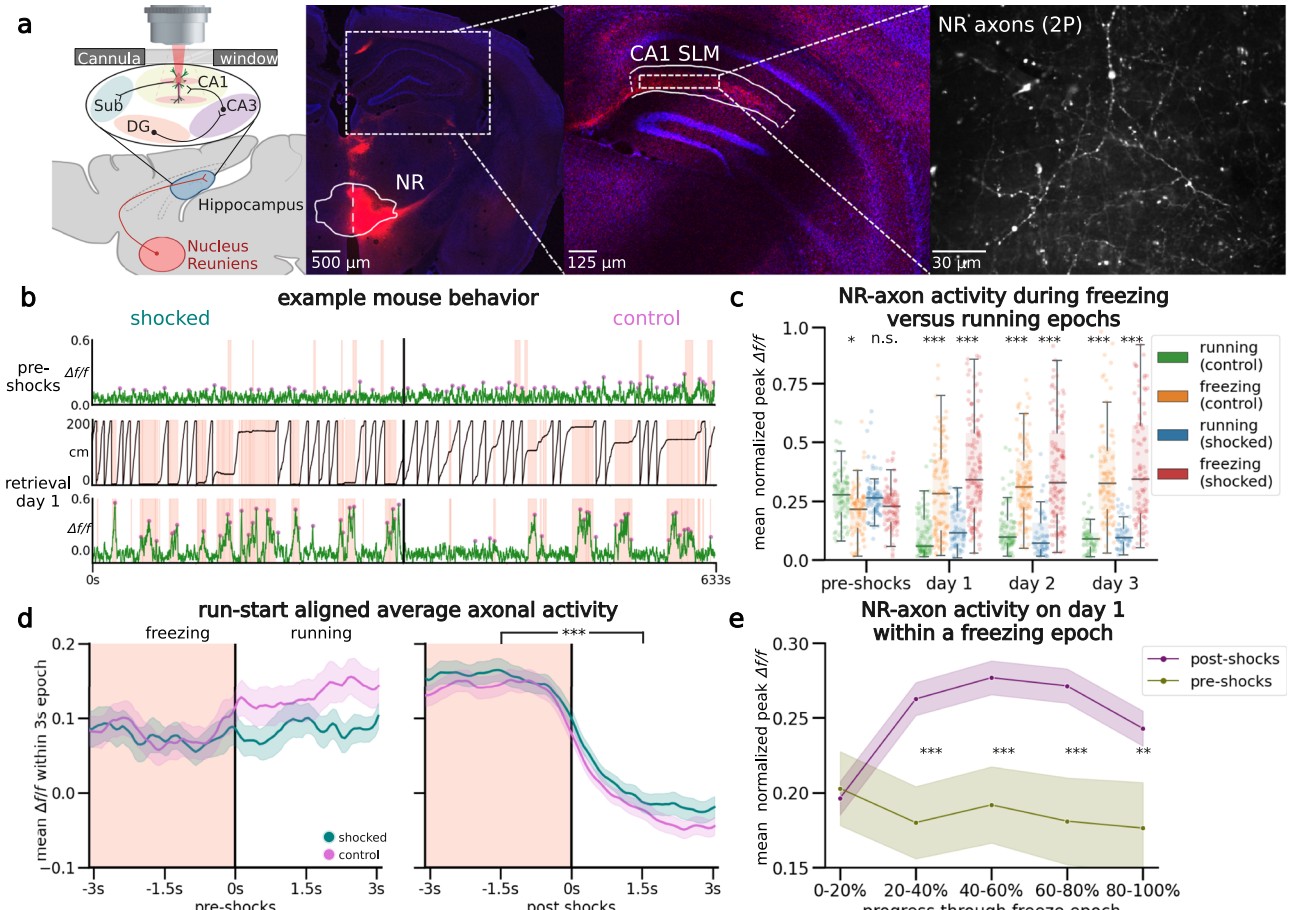

**Fig. 4 | Nucleus reuniens axons in CA1 tune to fearful freezing epochs following CFC. a** Left: schematic representation of NR axonal imaging. Mice were trained as in Fig. 1b. Middle left: NR mRuby expression in nucleus reuniens under confocal imaging. Middle right: axonal expression in the hippocampus limited to the SLM layer in subiculum and CA1. Right: example average FOV of NR axons in SLM through 2-photon during mouse behavior. **b** Example mouse NR axonal activity pre-shocks (Top) and on day 1 (Bottom) in the shocked (left) and control (right) contexts (total $N = 9$). Red shading indicates freezing epochs. Middle trace is the mouse position on retrieval day 1. **c** Normalized mean $\Delta f/f$ of axonal peaks per freezing epoch plotted as dots ($N = 927$), boxplot indicates median, 25-75th interquartile range, whiskers include all data points not determined to be outliers. In both contexts, mean normalized axonal activity increases post-shocks compared to pre-shocks, and remains elevated during post-shocks freezing epochs as compared to running epochs. Correspondingly, activity in running epochs decreased post-

shocks from pre-shocks (Two-sided student's $T$, $P$ control, shocked = pre-shocks: 0.09, 0.04, day 1: 9.42e-15, 6.27e-13, day 2: 6.64e-18, 7.19e-13, day 3: 1.22e-12, 1.47e-13). **d** Peach shading (left) indicates freezing epochs. Line indicates mean, shading indicates 95% CI. Pre-shocks, NR-CA1 axonal activity is comparable during freezing and running epochs. Post shocks, activity is significantly elevated during freezing when compared to running epochs (one-sided Wilcoxon rank sum left: $P = 0.14$, right: $P = 1.37e-52$). Freezing epochs displayed are 3–4 s long, additional epoch windows are shown in Supplementary Fig. 2e). **e** Normalized mean $\Delta f/f$ of axonal peaks were binned into five categories based on percent progress throughout the freezing epoch. Center lines indicate mean. Shaded areas indicate 95th confidence interval. For the majority of the pause (20–100% freeze progress) in both contexts (purple), axonal activity was significantly increased than activity before shocks (green) (one-sided Student's $T$, $P = 0-20\%$: 0.81, 20–40%: 1.71e-3, 40–60%: 9.75e-4, 60-80%: 1.46e-3, 80–100%: 8.70e-3). Created with BioRender.com.

## Discussion

Our findings expand on a previous canon of work that indicates both the mPFC-NR projection and NR itself are required for extinction of contextual fear and prevention of fear over-generalization[29,33–35]. Our results suggest that in addition to these roles, NR reduces time spent freezing following CFC by suppressing CFMR as it occurs during freezing epochs. We found that the NR-CA1 pathway is a key component of the circuit responsible for mediating the fear-suppressive function of NR. This is supported by our observation that NR axons in CA1 become selectively tuned to freezing epochs, 'ramping' up activity after the initiation of freezing epochs following CFC, and inhibiting the NR-CA1 pathway lengthens freezing epochs. The function of the NR-CA1 pathway in CFMR suppression is not restricted to the context in which shocks were presented, but extends to similar contexts where shocks never occurred. This seems to limit over-generalization as shown by NR-CA1 inhibition reducing context discrimination. Lastly, the process of suppressing ongoing CFMR by

the NR-CA1 pathway also has longer-term effects, as shown by reduced extinction on day 2 following NR-CA1 inhibition on day 1. In summary, our observations support a framework in which the NR-CA1 pathway actively suppresses fear responses by disrupting ongoing hippocampal-dependent CFMR to promote non-fearful behavior, and this process also limits fear over-generalization and promotes fear extinction. Supplementary Fig. 5 brings all our main findings together in a cartoon model.

Interestingly, we did not observe a significant difference in NR-axon activity during CFMR between contexts on retrieval days in our NR-axon recordings, despite NR-CA1 inactivation reducing discrimination between these contexts. It could be that while the NR-axons in CA1 are not contextually modulated, their activity induces postsynaptic dynamics in CA1 that encode differences in context. This is supported by previous work showing that CA1 is specifically necessary for the context-dependence of fear extinction[52]. We also found the difference between NR-axonal activity between freezing and running

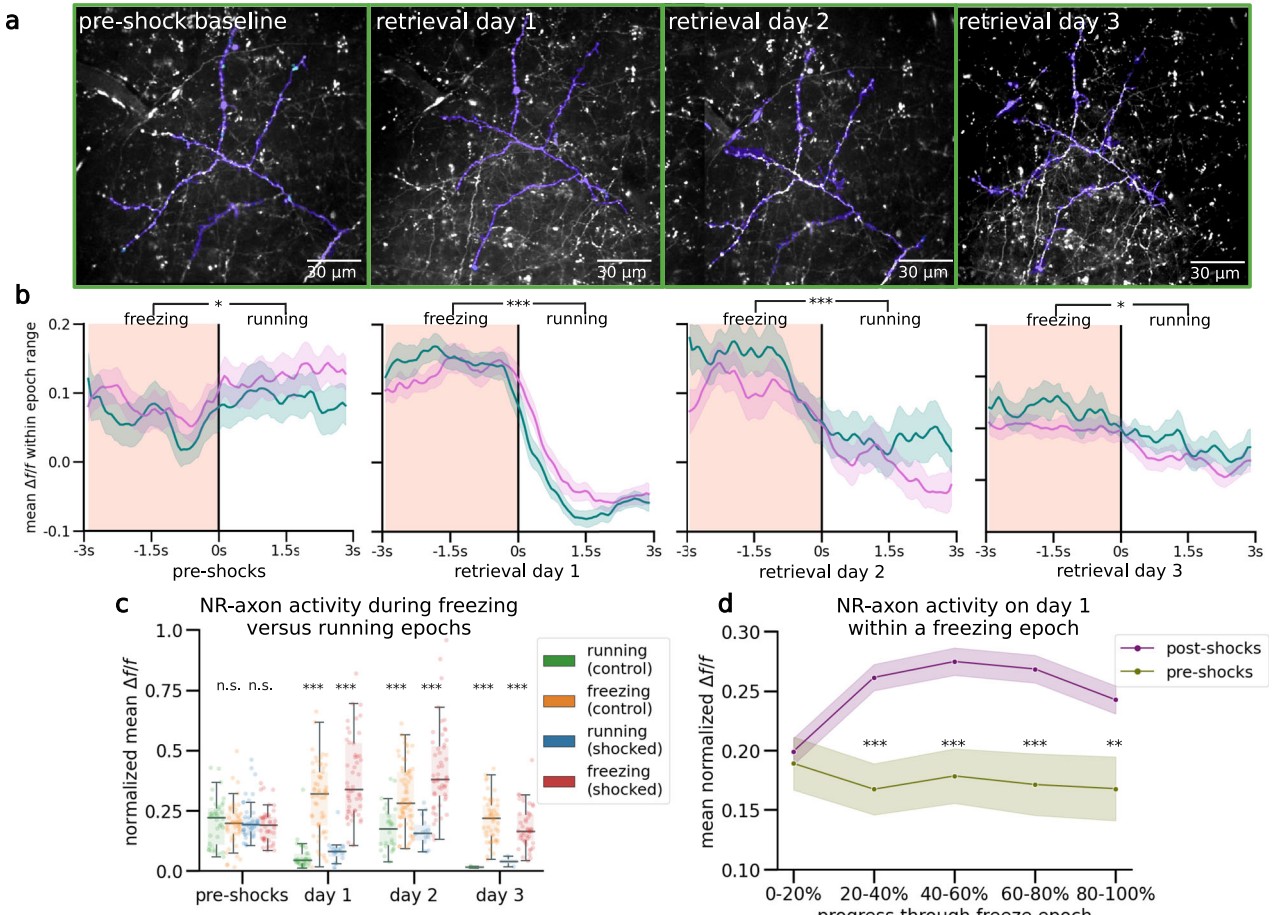

**Fig. 5 | Multi-day tracking of the same nr axons reveals fearful freezing tuning that decays with extinction. a** Example field-of-view (FOV) tracked across 4 days. FOVs are directly outputted from the Suite2p mean image with no color correction modifications applied, with Suite2p ROIs that comprised the final combined trace highlighted in purple on all four days, demonstrating our capacity to track the same NR-CA1 axonal structure over multiple days ($N = 4/10$ imaged mice). **b** Average axon activity of the 4 multi-day tracked axons. Peach shading (left side of each panel) indicates freezing epochs. Colored pink line indicates mean in control context and teal line indicates mean in shocked context, shading indicates 95% CI. Pre-shocks (far-left panel), NR-CA1 axonal activity is comparable during freezing and running epochs in both contexts. On post-shock retrieval days 1–3, activity is significantly

elevated during freezing compared to running epochs in both the shocked and control contexts (one-sided Wilcoxon rank sum, $P$ = pre-shocks: 0.33, day 1: 3.75e-8, day 2: 2.17e-5, day 3:0.02). **c** Analyses same as Fig. 4c, but on the subset of multi-day imaged axons (Two-sided students' $T$ test: shocked $P$ = pre-shocks: 0.21, day 1: 4.10e-3, day 2: 3.76e-8, day 3: 1.34e-6, control $P$ = pre-shocks: 0.59, retrieval day 1: 2.19e-13, retrieval day 2: 1.26e-5, retrieval day 3: 4.2e-4). **d** Analyses same as Fig. 2e, but on the subset of multi-day imaged axons. Center lines indicate mean. Shaded areas indicate 95th confidence interval. (Students' $T$ test $P$ = 0–20%: 0.67, 20–40%: 8.60e-6, 40–60%: 1.24e-4, 60–80%: 1.46e-4, 80–100%: 3.94e-3). Created with BioRender.com.

epochs following CFC (activity tuned to freezing epochs) did not decrease over days, even as mice decreased their time spent freezing. However, we did observe a decrease in axon activity tuned to freezing on retrieval day 3 in our multi-day tracked axon dataset, indicating that individual CA1-projecting NR neurons may become untuned to freezing epochs as extinction occurs, while population responses may not. Previous work shows that extinction does not erase previously learned contextual fear memories, as reactivation of hippocampal fear memories rapidly reinduces fear behavior[21,53]. This suggests that fear memories are retained but are dormant after extinction. Continued differential activity of NR-axons between freezing and running epochs in CA1, even after extinction, may be necessary to prevent the maladaptive retrieval of dormant fear memories, therefore enabling successful extinction learning.

A recent study showed that during freezing epochs in remote post-conditioning CFMR, optogenetic activation of NR significantly shortened freezing epochs, while inactivation lengthened freezing epochs[54]—in agreement with our NR-CA1 inhibition results. These authors revealed a transient increase in NR activity before the termination of freezing epochs, and showed a similar signal in the NR-

BLA (basolateral amygdala) pathway. The profile of the NR and NR-BLA activity during freezing epochs they report differs from the profile we report, as they ramped up at the end of a freezing epoch and remained high during running. What could be causing this discrepancy? One key difference is the time period in which the NR and NR-BLA signals occur, compared to our reported NR-CA1 signal. We recorded 1 day following CFC, whereas NR and NR-BLA signals were measured 30 days following CFC, a time period in which memories are considered remote and no longer dependent on the hippocampus[55]. In addition, the authors showed that the NR-BLA pathway was not necessary to facilitate extinction one-day following shocks, instead only being activated for remote memory retrieval. In our results, we show that the NR-CA1 pathway facilitates extinction across the timescale of a day. This suggests that CA1 and BLA projecting neurons may come from separate populations of NR neurons with different activity dynamics, a hypothesis supported by evidence that NR partially segregates into distinct sub-populations by projection region[56].

Another possibility for divergent results is that our axonal measurements are recording localized axonal spiking activity decoupled

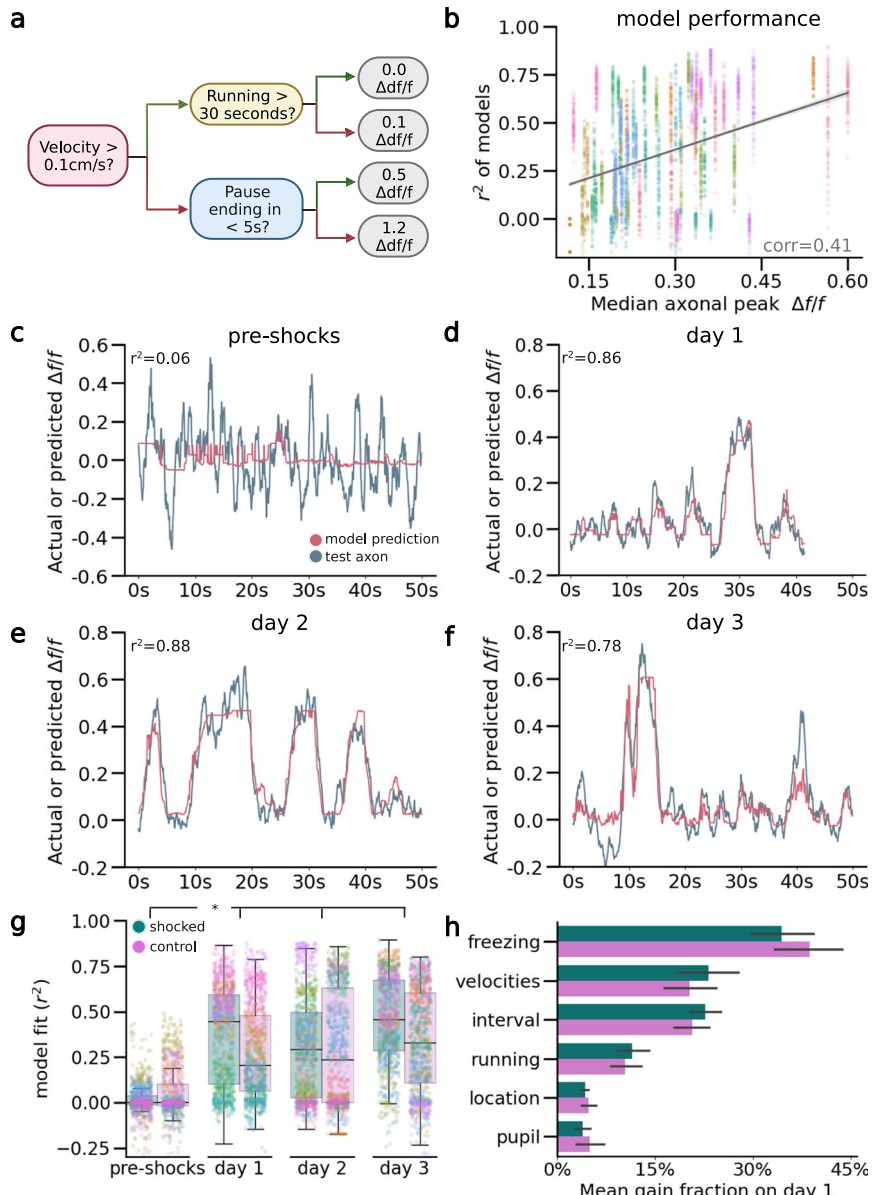

**Fig. 6 | Nucleus reuniens-CA1 axon activity is accurately predicted by a computational model following CFC. a** Simplified schematic of decision tree prediction. **b**, Model performed better on axons with higher fluorescence signals. Goodness of model fit $r^2$ was calculated for all model runs and plotted against median unnormalized axonal peak height as a proxy for data quality. Graded color dots are coded per mouse to visualize within-mouse stability ($N = 8000$ total runs; 10 mice, run 100 times per mouse, day, and context). **c–f** Examples of model prediction for the same axon in the same mouse tracked across days in the shocked context. Matched control context model examples are shown in Supplementary Fig. 4d). **g** Points indicate goodness of fit $r^2$ for each model run, color-coded by mouse, boxplot indicates median $r^2$, 25–75th interquartile range, whiskers include

all data points not determined to be outliers. Median model performance improved for both the shocked (pink) and control (teal) contexts across all days post-shocks, compared to pre-shock, in 8/10 imaged mice (two-sided Wilcoxon Rank Sum, $P =$ Shocked: day 1: 0.03, day 2: 0.02, day 3: 0.02, control: day 1: 0.04, day 2: 0.04, day 3: 0.03). **h** Gain fraction plotted per category across mice ($N = 8000$ total runs; 10 mice, run 100 times per mouse, day, and context), bar height equals mean, error bar indicates SEM. Model parameters pertaining to information about pausing, velocity, and duration of time paused or remaining in either a pausing or running interval ('interval') were used more than model parameters pertaining to running information, location on the track, or pupil information. Full gain fractions are shown in Supplementary Fig. 4b. Created with BioRender.com.

---

from somatic spiking activity, which has been observed in other circuits[57,58]. In this case, local synaptic modulation in CA1 may be exerting excitatory or inhibitory impacts on NR-axon activity, decoupling its activity profile in CA1 from somatic activity in NR. Future work using closed-loop optogenetic stimulation of the NR-CA1 pathway during freezing epochs, examining the existence or incidence of separate NR neurons projecting to BLA and CA1, investigating NR-CA1 activity at remote time points, and examining the activity of specifically NR-CA1-projecting somas is needed to directly test these hypotheses.

Interestingly, we found that after the NR-CA1 signal ramps up upon the start of freezing epochs it starts to ramp back down before the end of freezing epochs. We hypothesize that the ramping down at the end of freezing epochs could indicate a neural preparation or decision-making phase during which fear memory retrieval in the hippocampus has been suppressed but the decision to move has yet to occur. It also takes time for the brain to prime the motor circuits and coordinate the necessary movements before the actual initiation of movement. The CA1 is not a motor region, so the suppression of a fear memory in this region is unlikely to immediately cause

movement. The ramping down of the NR-CA1 signal may reflect the completion of fear memory suppression, allowing the animal to respond quickly when the decision to move is made, but the decision may take time. Animals may engage in an internal evaluation or computation of relevant information, such as assessing potential risks or benefits associated with the impending movement. The decision-making processes and the subsequent activation of motor commands may be occurring as the NR-CA1 signal ramps down having exerted memory suppression.

The input driving the NR-CA1 pathway is most likely from the mPFC, encompassing both the prelimbic (PL) and infralimbic (IL) regions. While PL is needed for fear acquisition and retrieval, IL is necessary for the opposing task of fear suppression and preventing over-generalization[59–62]. The likely opposing influences of IL and PL on NR during CFMR illustrates the importance of understanding NR output pathways. Our results indicate that a fear suppression signal circuit may be transmitted from IL, through NR, and into CA1 during CFMR. Of note, a small population of NR neurons that project both to CA1 and either PL or IL may have a key role in facilitating cross-regional theta and beta synchrony associated with hippocampal-dependent memory processing[28,63,64]. We cannot rule out that some of our recorded NR-axons collaterally project to mPFC, however, since this population makes up a small subset of all NR neurons (~3–9%[63]), we expect the majority of our recordings to be from non-dual projecting neurons. It additionally remains to be seen if the NR-CA1 exclusively projecting versus the NR-CA1 dual projecting populations have distinct dynamics during CFMR.

The amygdala, specifically the BLA, is a critical region that associates contextual information with fear[65,66]. Ultimately, short-term CFMR suppression and long-term extinction likely involves changing the contextual information sent out of the hippocampus to the BLA, either directly or indirectly, to an output not associated with fear in BLA. A key question that arises from our work is how the NR-CA1 pathway potentially disrupts CFMR-associated neural dynamics in CA1. NR exclusively projects to the SLM within CA1, where the distal dendritic tuft of pyramidal neurons receive targeted synaptic input from both medial and lateral EC and local inhibitory interneurons[44,67–69]. Whether NR directly synapses on these dendrites is under contention, with contradictory anatomical and electrophysiological reports supporting evidence for and against these direct synapses[29,50,70,71]. Electrophysiological stimulation of NR projections to CA1 in rodent slice work has largely supported that NR projections depolarize, but do not directly drive firing in pyramidal neurons[48–50,72], with one notable early exception[47].

Interestingly, NR and EC have been proposed to both project to the same dendritic compartments, and dual activation of NR and EC projections in slice amplifies nonlinear dendritic spiking, implying that NR/EC interactions may be important in vivo for synaptic plasticity[49,56]. Such dendritic-spike-induced plasticity has been associated with the formation of new place fields in novel environments[73] and could provide a mechanism through which NR both disrupts CFMR and promotes extinction learning. Additionally, either NR or EC projections to SLM, when coincident with CA3 inputs through schaffer collaterals, induce burst firing in CA1 pyramidal cells[74–77]. CA1 pyramidal cell bursts are also capable of inducing new place fields in CA1 through behavioral timescale synaptic plasticity (BTSP)[74,76–78]. If our newly reported NR input to CA1 pyramidal cell apical tuft dendrites during fearful freezing epochs coincides with CA3 inputs, their combined activity could induce burst firing and initiate BTSP. The bursts themselves could disrupt population dynamics to "jolt" the network out of CFMR, enabling the behavioral transition from freezing to running, while also inducing new place cell representations to form (remapping) through BTSP to support extinction learning[79].

This framework could explain why inhibiting the NR-CA1 pathway on retrieval day one reduced fear extinction on retrieval day 2. In effect, we may have prevented BTSP from inducing remapping and thus prevented extinction learning. Alternatively, reduced fear extinction on day 2 could be caused by the increased freezing behavior on day 1 during NR-CA1 inhibition, i.e., more freezing on day 1 leads to more freezing on day 2, although this effect was not observed in the control context. Whether delayed extinction on day 2 results from disruption to circuit-level processes on day 1 or from increases in freezing behavior on day 1 remains to be investigated. However, regardless of specific cause, our data support the conclusion that the NR-CA1 pathway is necessary for normal fear extinction.

NR could also disrupt CA1 dynamics through inhibition. It is well established that NR induces strong firing in various hippocampal interneuron populations with dendritic processes in SLM[47–50]. Which specific inhibitory populations are directly stimulated by NR is an open question, and one that has wildly divergent implications for the overall impact of NR on CA1 activity. In the case where NR-axons in SLM exclusively target inhibitory interneuron postsynaptic partners, the overall impact of NR-CA1 pathway activation on CA1 pyramidal population activity could still be net inhibitory, net excitatory, or selectively mixed. NR-axons could activate inhibitory micro-circuits that disrupt awake replay of location-specific activity sequences of the shocked context during freezing[80], or silence temporally-restricted reactivation of engram cells[21] to induce fear memory suppression and enable extinction learning. Further research on the impact of NR on CA1 dendritic and somatic population dynamics is needed to unravel how the NR-CA1 pathway mechanistically induces suppression of ongoing CFMR.

## Methods

### Experimental model and subject details

All experimental and surgical procedures were in accordance with the University of Chicago Animal Care and Use Committee guidelines. We used 10–20 week old male C57BL/6 J wildtype (WT) mice (23–33 g). Male mice were used over female mice due to the size and weight of the headplates (9.1 mm × 31.7 mm, ~2 g) which were difficult to firmly attach on smaller female skulls, and low weights reached under water restriction in female mice making the additional weight of the tailcoat potentially burdensome and interfere with experimental results. Mice were individually housed in a reverse 12 hour light/dark cycle and behavioral experiments were conducted during the animal's dark cycle. We are unaware of any influence of strain or sex on the parameters analyzed in this study. A total of 79 mice were used, 46 of which were used for data analysis shown in this paper. 33 mice were excluded for not meeting running behavior criteria (See Methods: Behavior). Of these, 20 never reached the 4 traversals/minute cutoff after 14+ days of training, and 13 did not meet movement criterion after removal of water reward and addition of the tailcoat. Of the 46 mice that did meet behavioral criteria, 20 mice were used in the NR-CA1 intact group, 10 of which were imaged. Of the remaining 10, although they had imaging windows implanted, 6 did not have sufficient signal-to-noise on a pre-experimental day to image through CFC, and 4 imaging recordings were eliminated *post hoc* for z-motion drift. In the remaining 26 mice, nine mice were used for NR-CA1 DREADD inhibition and 17 mice were used as controls: 6 mice were used for no-shock control, 4 mice were used for mCherry DREADD control, and 7 mice were used for saline DREADD control.

### Mouse surgery and viral injections

Mice were anesthetized (~1–2% isoflurane) and injected with 0.5 ml of saline (intraperitoneal IP injection) and 0.5 ml of Meloxicam (1–2 mg/kg, subcutaneous injection) before being weighed and mounted onto a stereotaxic surgical station (David Kopf Instruments). A small craniotomy (1–1.5 mm diameter) was made over the hippocampus (±1.7 mm lateral, −2.3 mm caudal of Bregma) or NRs (0.0 lateral, −0.6 caudal of Bregma). For NR imaging experiments, an axon-targeted

genetically-encoded calcium indicator, AAV9-axon-GCaMP6s-P2A-mRuby3 (pAAV-hSynapsin1-axon-GCaMP6s-P2A-mRuby3 was a gift from Lin Tian Addgene viral prep # 112005-AAV9; http://n2t.net/addgene:112005; RRID:Addgene_112005) was injected (~50 nL at a depth of 4.1 mm below the surface of the dura) using a beveled glass micropipette leading to GCaMP6s and mRuby expression in a population of NR neurons. For DREADD experiments, first AAVrg-hSyn-DIO-hM4D(Gi)-mCherry (pAAV-hSyn-DIO-hM4D(Gi)-mCherry was a gift from Bryan Roth Addgene viral prep # 44362-AAVrg; RRID:Addgene_44362) was injected into bilateral hippocampal CA1 SLM (~50 nL per side at a depth of −1.5 mm below the surface of the dura). In the same surgical procedure, AAV9-hSyn-Cre (pENN.AAV.hSyn.-Cre.WPRE.hGH was a gift from James M. Wilson Addgene viral prep # 105553-AAV9; http://n2t.net/addgene:105553; RRID:Addgene_105553) was injected into bilateral NR (~100 nL at a depth of −4.1 mm). For NR-DREADD Controls, AAVrg-hSyn-DIO-mCherry (pAAV-hSyn-DIO-mCherry was a gift from Bryan Roth Addgene viral prep # 50459-AAVrg; http://n2t.net/addgene:50459; RRID:Addgene_50459) was substituted for AAVrg-hSyn-DIO-hM4D(Gi)-mCherry. See: Table 1: Key Resources Table for details. Afterwards, the site was covered using dental cement (Metabond, Parkell Corporation), and a metal head-plate (9.1 mm × 31.7 mm, Atlas Tool and Die Works) was also attached to the skull with the cement. Mice were separated into individual cages and water restriction began the following day (0.8–1.0 ml per day). At least 7 days following injection surgery, and approximately 7 days prior to the beginning of mouse training, mice underwent another surgery to implant a hippocampal window as previously described[81]. Following implantation, the head-plate was reattached with the addition of a head-ring cemented on top of the head-plate which was used to house the microscope objective and block out ambient light. Post-surgery mice were given 1–2 ml of water/day for 3 days to enhance recovery before returning to the reduced water schedule (0.8–1.0 ml/day). Expression of axon-GCaMP6s reached a steady state ~50 days after the virus was injected, as monitored through 2p imaging. Expression of hM4D(Gi)-mCherry was validated using post hoc confocal imaging.

## Behavior

Our virtual reality (VR) and treadmill setup were designed similarly to previously described setups[40]. The virtual environments that the mice navigated through were created using VIRMEn[82]. Mice were head-restrained with their limbs comfortably resting on a freely rotating styrofoam wheel ('treadmill'). Movement of the wheel caused movement in VR by using a rotary encoder to detect treadmill rotations and feed this information into our VR computer, as in (Heys et al., 2014; Sheffield et al., 2017). During training, mice received a water reward (4 μl) through a waterspout upon completing each traversal of the track (a lap), which was then associated with a clicking sound from the solenoid. Upon receiving the water reward, a short VR pause of 1.5 s was implemented to allow for water consumption and to help distinguish traversals from one another rather than them being continuous. Mice were then virtually teleported back to the beginning of the track and could begin a new traversal. Mice were also teleported to the beginning of a new contextual exposure.

Four distinct VR contexts were used in this experiment, and all contexts can be found on GitHub at https://github.com/hmacomber/NR-Analysis. The training VR has local cues of a black track with gray squares at regular intervals, and gray walls with black ovals. The end of the track has a gray wall with neon green dots. Additional local cues include a green and white checkered overhead 'bridge' and a black flag with a white star towards the end of the track. Distal cues include black and gray rectangles, and a cylindrical gray tube with lighter checkered pattern on an overhead oval. The first of the two VRs used as shocked or control contexts has a light gray track with four white diamonds for the first half of the track and a black track with two white lines for the second, black

walls with white lines and gray dots at regular intervals, and bright solid green end wall with a black line in the middle. Additional distal cues are provided by a gray and black chevron pattern sky, neon green walls with X patterns in white, black, and neon blue cylinders in the later half of the track, and a black conical with white stripes at the end of the track. The second of the two VRs has a black track, surrounded by white lines and small gray dots. Green and yellow conical structures with white stripes dot the landscape. The far end of the track has a striped wall.

Mouse behaviors (running velocity, track position) were collected using a PicoScope Oscilloscope (PICO4824, Pico Technology). Pupil tracking was done through the imaging software (Scanbox, Neurolabware) at 15.49 Hz, using Allied Vision Mako U-130b camera with a 25 mm lens and a 750 nm longpass IR filter. IR illumination from the objective was used to illuminate the pupil for tracking. Behavioral training to navigate the virtual environment began ~7 days after window implantation (~30 minutes per day) and continued until mice reached a speed of greater than four traversals per minute, which took 10-14 days (although some mice never reached this level). This high level of training was necessary to ensure mice continued to traverse the track similarly after reward was removed. Initial experiments showed that mice that failed to reach this criterion typically would not traverse the track as consistently without reward[40] a potential confound for post-shocks freezing data (data not shown). Mice that did not reach this criterion were not used for these experiments (28 mice removed across all conditions).

## Contextual fear conditioning paradigm

For mice that reached criteria in the training environment (>4 traversals per minute), mice were first exposed to two novel environments without water reward for 322 s (~5 minutes) each, with the addition of a custom-made tailcoat made of conductive fabric (Adafruit). Only mice that continued to maintain a speed of 4 traversals >minute without water rewards and with the tailcoat on were allowed to continue the experiment. Subselecting for mice with this consistent running behavior helped us to ensure that freezing responses recorded later were not due to the presence of the tailcoat or any discomfort from head fixation or removal of reward. Here onwards, the tailcoat was kept on the mouse during the experimental sessions on all subsequent experimental days. Each contextual exposure was for a duration of ~5 minutes. Prior to experimental day 0, mA level of shock delivery was confirmed using an oscilloscope. On day 0, mice were exposed to both novel contexts, then shocked in one of the two contexts, administering 6 × 0.6 mA shocks delivered at an interval of 20–26 seconds each, (Coulbourn Instruments Precision Animal Shocker). Mice displayed rapid sprinting behavior when they received the tail shock, allowing us to confirm the delivery of shocks in real-time (Fig. 1e). On subsequent days, mice were exposed to both the shocked and non-shocked (control) contexts pseudorandomly, for 3 days.

## DREADD experimental protocol

To activate the hM4D(Gi) receptor and silence a subset of NR glutamatergic neurons that project to CA1, we used Deschloroclozapine dihydrochloride (DCZ, MedChemExpress). DCZ was chosen over CNO as an h4MDi agonist as DCZ has a significantly increased potency, therefore enabling a 100-fold dosage reduction, heightened selectivity for h4MDi receptors over endogenous receptors, and a significantly more rapid onset than CNO. In addition, the DCZ metabolites C21 and DCZ- N-oxide are reported at negligible concentrations (<0.2 nM), compared to the known tendency of CNO to metabolize to Clozapine at higher concentrations, a compound with significant off-target receptor binding in the mammalian brain[43]. Because of these factors, we chose to use DCZ for inactivation, as in our past work[40].

Once mice met training criteria, they were habituated to the injection process. They were exposed to the rewarded training environment for ~10 min. Afterwards, they were removed from the VR setup, placed in the holding room, and injected with ~150 μL of a 12% DMSO/Saline solution. After ~30–45 min, they were placed back in the VR setup and exposed to the rewarded training environment again for an additional 10 min. This was repeated for 3 days to acclimate mice to the injection procedure. Mice additionally received ~150 μL of a 12% DMSO/Saline solution on Day −1 of the experiment 30 minutes prior to first exposure to both neutral contexts to mimic conditions on Day 0.

For animals receiving DCZ injections, i.e., both the experimental NR-CA1 inhibited AAVrg-hSyn-DIO-hM4D(Gi)-mCherry group and the control NR-CA1 intact AAVrg-hSyn-DIO-mCherry group, DCZ was dissolved in DMSO at .02 mg/mL concentration and stored at −80 °C on day 0. On retrieval day 1, DCZ solutions were thawed to room temperature and diluted to 0.01 mg/mL with DMSO/Saline. ~30 minutes prior to context exposure, mice were brought to a holding room and IP injected with 0.1 mg/kg DCZ of a .02 mg/mL solution. A separate control group with hM4Di expression intact received DMSO/saline instead of DCZ on retrieval day 1. These mice were injected with a weight-matched quantity (~100–150 μL) of saline in place of 0.01 mg/mL DCZ. In all groups, a quantity of DMSO/Saline solution identical to IP injection amount on Day 1 (~100–150 μL) was injected on all other experimental days, ~30 minutes prior to imaging, to control for the impact of any potential IP injection-induced stress. Imaging protocol for all DREADD NR-CA1 inhibited experimental mice and intact controls was kept identical to VR-CFC NR-axon imaged mice, with the addition of a 'dark' imaging session after context exposures of the same duration, where no context was displayed on screens, for ~5 minutes, to check for any impact of DCZ on movement (Fig. 3f).

## Two-photon imaging

Imaging was done using a laser scanning two-photon microscope (Neurolabware). Using a 8 kHz resonant scanner, images were collected at a frame rate of 15.49 Hz with unidirectional scanning through a 16×/0.8 NA/3 mm WD water immersion objective (MRP07220, Nikon). axon-GCaMP6s was excited at 920 nm and mRuby was excited at 1040 nm with a femtosecond-pulsed two-photon laser (Insight DS +Dual, Spectra-Physics) and emitted fluorescence was collected using two GaAsP PMTs (H11706, Hamamatsu). The average power of the laser measured after the objective ranged between 60–100 mW, and was kept constant across days of imaging. A single imaging FOV was positioned between 350–500 μm below the putative surface and 400–700 μm equally in the x/y direction to collect data from as many NR axonal segments as possible. Time-series images were collected through Scanbox (Neurolabware) and the PicoScope Oscilloscope was used to synchronize frame acquisition timing with behavior. When possible, the same axonal field was returned across days (Fig. 5, $N = 4/10$ imaged mice).

## Immunohistochemistry and confocal imaging

Expression of either hm4D(Gi)-mCherry or GCaMP6s-mRuby in glutamatergic neurons in NR were checked post hoc. Mice were anesthetized with isoflurane and perfused with ~10 ml phosphate-buffered saline (PBS) followed by ~20 ml 4% paraformaldehyde in PBS. Brains were removed and immersed in 30% sucrose solution overnight before being sectioned at 30 μm-thickness on a cryostat. Brain slices were collected into well plates containing PBS. Slices were washed five times with PBS for 5 min then were blocked in 1% Bovine Serum Albumin, 10% Normal goat serum, 0.1% Triton X-100 for 2 hrs. Brain slices were then incubated with either 1:500 rabbit-α-mCherry (ab167453, Abcam) or 1:500 goat-α-mRuby (STJ140251, St John's Laboratory) in a blocking solution at 4 °C. After 48 hrs, the slices were incubated with either 1:1000 goat-α-rabbit Alexa Fluor

488 secondary antibody (A32731, ThermoFisher) or 1:1000 rabbit-a-goat Alexa Fluor 488 secondary antibody (A27012, ThermoFisher) respectively, for 2 hrs. Brain slices were then collected on glass slides and mounted with a mounting media with DAPI (SouthernBiotech DAPI-Fluoromount-G Clear Mounting Media, 010020). Whole-brain slices were imaged under ×10 and ×40 with a Caliber I.D. RS-G4 Large Format Laser Scanning Confocal microscope from the Integrated Light Microscopy Core at the University of Chicago.

## Image processing and ROI selection

Time-series images were preprocessed using Suite2p (Pachitariu et al., 2017). Movement artifacts were removed using rigid and non-rigid transformations and assessed to ensure the absence of drifts in the z-direction. Datasets with visible z-drift were discarded ($N = 4$). All datasets collected during shock administration on Day 0 were discarded, due to the high-velocity post-shocks sprinting behavior of mice making FOVs too unstable for reliable analysis. Regions of interest (ROIs) were also defined using Suite2p (Fig. 5a) and manually inspected for accuracy. Baseline corrected Δf/f traces across time were then generated for each ROI.

In addition, to control for in-experiment motion artifacts for small axonal segments, a red mRuby channel was recorded simultaneously to GCaMP6s channel recordings. Per ROI, a savitzky-golay filter was applied to both channels to smooth the signal. Then, the demeaned red channel was 'subtracted' from the demeaned green channel, by orthogonalizing their vectors in variance space. That is, we took the projection of the red channel onto the green channel as $\frac{Cov(Green,Red)}{Cov(Red,Red)} \cdot Red$, and then subtracted that vector from the green channel. This results in a new vector which is guaranteed to have zero covariance with the red channel, thus removing any linear effects of the background fluorescence on the trace. All ROIs were analyzed for covariance, and any ROIs exceeding the 99th percentile of a shuffle distribution were combined using PCA and the first PC taken, in a method similar to Kaufman et al.[83] To ensure traces had sufficient activity for analysis, all mice used were required to have one axon per FOV with activity that exceeded 10% Δf/f twice on each experimental day ($N = 10$ mice). The activity of each axon was then internally rescaled per day to the 99th percentile of max activity to account for inter-axonal differences in calcium brightness. Peaks were calculated using the scipy.signal.find_peaks package with a required minimum height of 10% Δf/f, distance of 0.5 s, and prominence of 0.1. One axon per mouse was selected in each FOV for numerous reasons. In our dataset, NR axons in CA1 are highly branched. In most FOVs, a single axon branched visibly across the majority of the FOV, and was determined to be visually connected as well as having highly correlated activity among ROI segments. In the majority of cases, ROI segments with detectable activity determined by Suite2p were correlated with one another above an $r^2$ of >0.2. Because of this relatively high correlation among identified segments, we could not rule out that all segments in each FOV originated from the same neuron in NR, even when they visually did not connect, and therefore we were conservative with our axonal selection to visually connected segments above our ROI correlation threshold, and only these ROIs were considered to be from the same axon. Further, only ROIs in a FOV exceeding the 99th percentile of a shuffle distribution were combined. We did the combining using PCA, and took the first PC, the remaining ROIs were discarded.

## Pupil measures

To obtain images with dark pupils and high contrast around the borders of the pupils, pupil images were inverted, and their brightness/contrast was adjusted in ImageJ. Pupil area, pupil center of mass (COM), Pupil x and y positions, and blinking area were obtained using FaceMap (Stringer et al. 2019). Pupil data during

blinking periods (frames where blinking area <mean − twice the standard deviation of the blinking area) was removed and the pupil data was interpolated to match the two-photon imaging frame rate (15.49 Hz). Pupil area and x and y position data were smoothed with a savitzky–golay filter.

## Boosted trees model

The encoding model used is the python implementation of the open-source gradient-boosted trees algorithm XGBoost[51]. Behavioral model parameters (described below) were used to predict axon trace values. For reproducibility, the seed was set to 42. Data were then split into laps, and split using an 80/20 train/test regime. Model was run either per mouse (Fig. 6) or across mice (Supplementary Fig. 4), per, day, and context paradigm, for a total of 8000 runs ($N = 10$ mice, 4 days, 2 contexts, 100 draws). Chance performance was determined by shuffling neural activity by traversal compared to behavioral readout per mouse, across contexts and days. Model hyperparameters were set to: gamma = 1, learning_rate = 0.01, n_estimators = 1000, base_score = 1, early_stopping_rounds = 5. The coefficient of determination $r^2$ is defined as $1 - \frac{u}{v}$ where $u$ is the residual sum of squares $\sum (y_{true} - y_{pred})^2$ and $v$ is the total sum of squares $\sum (y_{true} - y_{mean})^2$. The best possible $r^2$ score is 1.0, and the $r^2$ score can be negative because the model can be arbitrarily worse than chance. For ease of interpretability in Fig. 6h, the following groupings of related behavioral variables were made and the mean contribution found within each group: freezing = ('freeze', 'is freezing', 'freeze remaining', 'is postfreeze', 'freeze progress', 'freeze elapsed'), velocities = ('recorded velocity', 'velocity back 15 frames', 'velocity back 8 frames', 'velocity forward 8 frames', 'velocity forward 15 frames'), running = ('is running', 'running progress','running remaining', 'is backtracking','running_elapsed', interval = ('interval_elapsed', 'interval_remaining','interval_progress', location = 'location', pupil = ('pupil area', 'pupil x position', 'pupil y position'). We used the importance type 'gain' parameter to determine the importance of each figure to the model's overall performance. 'Gain' is how much an individual feature contributed to model accuracy (i.e., the distance between predicted and actual $r^2$ values) on each branch. For each feature's use in the model, that value is summed, then the median was taken across all models by context. Full gain fractions for each parameter are shown in Supplementary Fig. 4b.

## Behavioral parameters

All parameters described below were calculated per mouse, day, and context, and used in model training, with the exceptions of total displacement and shocks.

**Time to complete a traversal.** This was calculated as the total time (in seconds) taken by the animal to run from 0 to 200 cm. Frames recorded within the teleportation window were dropped from the analysis.

**Total displacement.** Total displacement was calculated as the distance traversed per mouse, per context, per day.

**Freezing.** Freezing epochs were determined as uninterrupted epochs where mouse velocity fell below 0.001 cm/s for at least 12 consecutive frames (-0.75 s). All epochs of velocity below 0.001 cm/s but not reaching 12 consecutive frames were not considered freezing or running, and were discarded from future analysis. Freezing epochs were then counted up, and each not in a freezing epoch assigned a '0', while each frame in a freezing epoch given a numeric value corresponding to the number of epochs in that recording (i.e., all frames that contained the 4th freeze of the recording would be assigned the integer '4'). Subsequent freeze features were then calculated, including the binary variable 'is freezing' which assigns a 1 to frames considered freezing, and 0 to

frames not considered freezing, two sawtooth functions 'freeze remaining', and 'freeze elapsed', which counts the frames from the beginning of a freeze up or down until the end of a freeze, respectively, and 'freeze progress' which tracks the progress of a freeze as a fraction from 0 to 1.

**Running.** Running was determined as any epoch where forward progress velocity was sustained over 0.001 cm/s for 2 consecutive frames. The variables 'is running', 'running remaining', 'running elapsed', and 'running progress' are calculated using the running epoch data in the same fashion as their freezing counterparts.

**Backward movement.** Some mice demonstrated backward movement behavior in the virtual environment post-shocks, where they made backwards movement through the context. This behavior was analyzed separately from running or pausing in Supplementary Fig. 1e. The binary variable 'is backtracking' assigns a 1 to frames considered backtracking, and 0 to frames not considered backtracking.

**Shocks.** Shock delivery was recorded through the Picoscope. Shock location on track and stereotyped post-shocks sprinting behaviors are quantified in Fig. 1c, h.

**Velocity.** Velocity was both directly measured through the picoscope encoder, and recalculated from position, to assess for accuracy. Recorded velocity was used for all velocity calculations and model training. Values were converted into cm/s for presentation.

**Velocity offsets.** Future and past velocity at -1 s and -0.5 s were calculated by offsetting the velocity to frames. The resulting non-existent 8 or 15 velocity frames at the beginning or end of the trace were extrapolated from the prior 15 frames.

**Acceleration.** Acceleration was calculated as the first derivative of recorded velocity.

**Intervals.** Three variables, 'interval elapsed', 'interval progress', and 'interval remaining combine pausing and running information into one datastream. Interval elapsed takes the component parts 'freeze elapsed' and 'running elapsed', and counts the time elapsed in either a pausing or running interval, before resetting at a switch point. 'Interval remaining' and interval progress do the same calculation, but using freeze remaining'/'running remaining' and 'freeze progress'/'running progress'

**Location.** Animal's position on virtual track was determined for each frame, and binned in 1 cm bins along the virtual track.

**Pupil Area.** Pupil area was calculated by FaceMap as previously described, then filtered with a savitzky-golay filter for smoothing.

**Pupil horizontal (x) movement.** Pupil x movement was calculated by FaceMap as previously described then filtered with a savitzky-golay filter for smoothing.

**Pupil vertical (y) movement.** Pupil y movement was calculated by FaceMap as previously described, then filtered with a savitzky-golay filter for smoothing.

## Statistics

For all data distributions, a Shapiro–Wilk test was performed to verify the data was normally distributed before undergoing further statistical tests. For behavioral calculations of percent time spent freezing between contexts (shocked, control), conditions

**Table 1 | Key resources table**

| Reagent or resource | Source | Identifier |
|---|---|---|
| Bacterial and virus strains | | |
| pENN.AAV9.axon.GCaMP6.P2A.mRuby3 | Broussard et al.[85] | Addgene #112005-AAV9' |
| pENN.AAVrg.hSyn.DIO.hM4D(Gi).mCherry | Krashes et al.[86] | Addgene # 44362-AAVrg |
| pENN.AAV.hSyn.Cre.WPRE.hGH | Wilson Lab Plasmids (unpublished) | Addgene # 105553-AAV9 |
| pENN.pAAV.hSyn.DIO.mCherry | Roth lab DREADDs (unpublished) | Addgene # 50459-AAVrg; |
| Experimental models: organisms/strains | | |
| Mouse: C57BL/6 J | Jackson Laboratories | JAX 000664 - C57BL/6J |
| Antibodies | | |
| rabbit-α-mCherry | Abcam | ab167453 |
| Goat-α-mRuby | St. John's Laboratory | STJ140251 |
| Goat-α-rabbit Alexa Fluor 488 | ThermoFisher | A32731, ThermoFisher |
| Rabbit-a-goat Alexa Fluor 488 | ThermoFisher | A27012, ThermoFisher |
| Software and algorithms | | |
| Fiji | Schindelin et al.[87] | https://imagej.net/software/fiji/; RRID:SCR_002285 |
| Suite2p | Pachitariu et al.[88] | https://github.com/MouseLand/suite2p |
| MATLAB | MATLAB. (2018).9.7.0.1190202 (R2018a). | https://www.mathworks.com/help/matlab/release-notes-R2018a.html |
| Python | Python 3.10.8 | https://www.python.org/downloads/release/python-3108/ |
| Pandas | Pandas 1.1.4 | https://pandas.pydata.org/ |
| XGBoost | XGBoost 1.5.0 | https://xgboost.readthedocs.io/en/stable/python/python_api.html |
| SciPy | SciPy 1.9.3 | https://scipy.org/install/ |
| Seaborn | 0.12.0 | https://seaborn.pydata.org/installing.html |
| R | R-4.3.1 | https://www.r-project.org/ |

(NR-intact, NR-inhibited, unshocked), and days (pre-shock, retrieval day 1, retrieval day 2, retrieval day 3) a three-way repeated measures ANOVA in R was conducted using the afex package. A similar ANOVA structure was created for the behavior in the subset of mice that were also imaged, and a two-way repeated measures ANOVA was constructed for the behavior of mice in the dark context, which could not be incorporated into the larger behavioral ANOVA due to the unbalanced experimental design, i.e., mice were not exposed to the dark context on the pre-shock day. Two-sided pairwise comparisons of estimated marginal means were then conducted on model outputs in R using the emmeans package, with a bonferroni multiple comparisons adjustment applied post hoc for a total of 51 tests. Any $p$ value corrected above 1 was reported as 1. Mann–Whitney $U$ tests were used to compare distributions displayed as histograms. For axonal data, a paired Wilcoxon signed rank test, unpaired Student's $T$ test, or an unpaired Mann–Whitney $U$ test was used. For samples with five data points or less, only a non-parametric test was used. Box and whisker plots were used to display data distributions where applicable. The box in the box and whisker plots represent the first quartile (25th percentile) to the third quartile (75th percentile) of the distribution, showing the interquartile range (IQR) of the distribution. The black line across the box is the median (50th percentile) of the data distribution. The whiskers extend to 1.5*IQR on either side of the box. A data point was considered an outlier if it was outside the whiskers or $1.5 \times$ IQR. Significance tests were performed with and without outliers. Data distributions were considered statistically significant only if they passed significance ($p < 0.05$) both with and without outliers. Significance numbers reported are without outliers. To model the probability distribution in the datasets and get an accurate idea of the data shape, a kernel density estimate was fitted to the data distribution and is shown alongside histograms. Cumulative probability distribution functions were compared using a Kolmogrov–Smirnov test. Correlations were performed using Pearson's correlation coefficient. $p < 0.05$ was chosen to indicate

statistical significance and $p$ values presented in figures are as follows: *$p < 0.05$, **$p < 0.01$, ***$p < 0.001$, N.S. not significant. Darker lines in the center of line plots are the mean, and shading is the 95% confidence interval, unless stated otherwise in text or figure legends. All regression analysis was conducted using the statsmodels Robust Linear Model package, which estimates a robust linear model via iteratively reweighted least squares, given a robust criterion estimator. The M-estimator minimizes the function $Q(e_i,\rho) = \sum \rho(\frac{e_i}{s})$ where $\rho$ is a symmetric function of the residuals and $s$ is an estimate of scale. We used standardized median absolute deviation for $s$ and Huber's loss function, as it is less sensitive to outliers. Shading on regressions indicates 95% CI. (see https://www.statsmodels.org/dev/examples/index.html#robust-regression for additional details). Some data preprocessing was done with MATLAB (Mathworks, Version R2018a). All other data and statistical analyses were conducted in Python 3.7.4, with primary data accrued in Pandas DataFrames, and data figures were made in Python 3.7.4 using the Seaborn and Matplotlib packages (https://www.python.org/). Schematic figures (Fig. 1a, b. Fig. 4a, Fig. 6a), some figure text, and figure layouts were made with BioRender (https://biorender.com/).

**Reporting summary**

Further information on research design is available in the Nature Portfolio Reporting Summary linked to this article.

## Data availability

The data generated in this study have been deposited in the github database under accession code https://doi.org/10.5281/zenodo.8393380[84]. Raw imaging data are available under restricted access due to their substantial size on the multi-terabyte scale. Access can be obtained by contacting the corresponding author. The processed imaging and behavioral data are available at the previous link. The relevant data generated in this study are provided in the Supplementary Information/Source Data file. Source data are provided in this paper.

## Code availability

The original code used to create figures from preprocessed data is available on GitHub at https://doi.org/10.5281/zenodo.8393380.

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

## Acknowledgements

This work was supported by The Whitehall Foundation, The Searle Scholars Program, The Sloan Foundation, The University of Chicago Institute for Neuroscience start-up funds, a New Innovator grant from the National Institutes of Health (1DP2NS111657-01) awarded to M.S., and a T32 training grant (T32DA043469) from National Institute on Drug Abuse awarded to S.K. We thank the University of Chicago imaging core for assistance with confocal imaging, and the University of Chicago animal care staff for ensuring the well-being of experimental animals. We thank Chad Heer for early help with imaging protocols. We thank Valerie Barreto and Cherry Wang for assistance in training animals, post hoc perfusion, immunohistochemistry, and confocal imaging. We thank Timothy Ratigan for their assistance with data analysis. We thank Antoine Madar for their assistance with statistical methodology. We thank Rossten Rad for their discussions on experimental design. Finally, we thank Douglas Goodsmith, Jim Heys, Timothy Ratigan, and Rossten Rad for their invaluable comments on previous versions of the manuscript.

## Author contributions

S.K. and M.S. conceived, designed, and tested the VR-CFC protocol. H.R. modified the VR-CFC protocol in collaboration with S.K. H.R. and M.S. conceived and designed the experiments. H.R. performed surgeries. H.R. and S.S. collected all in vivo behavioral and imaging data. H.R. and S.S. collected *post hoc* data. H.R. wrote the analysis code and analyzed all the data. H.R. and M.S. interpreted the data and wrote the manuscript, with significant contributions from S.K.

## Competing interests

The authors declare no competing interests.
