## [Peer Review File · Nature Communications]

A Thalamic-Hippocampal CA1 Signal for Contextual Fear Memory Suppression, Extinction, and DiscriminationREVIEWER COMMENTS

Reviewer #1 (Remarks to the Author):

This is an exciting paper examining the role for nucleus reuniens projections to hippocampal area CA1 in head-fixed, behaving mice. The authors show in a novel VR task performed on a treadmill that mice exhibit contextual fear conditioning which is manifest in discriminated conditioned freezing in the shock vs no-shock contexts. Chemogenetic inhibition of RE neurons projecting to dorsal hippocampus increased freezing in both the shock and no-shock contexts suggesting this pathway plays a role in inhibiting the expression of hippocampus-dependent fear memory. These effects were not simply due to nonspecific increases in freezing (or decreases in locomotor activity) and are similar to findings previously reported in rats. The authors also used 2P calcium imaging to record from RE terminals in CA1 SLM using axon-targeted GaMP6. These recordings revealed that RE terminal activity became tuned to bouts of freezing after conditioning, exhibiting greater activity during bouts of freezing than locomotion. Ultimately, the data suggest that RE projections to CA1 inhibit hippocampal fear memory and facilitate discrimination and extinction. Altogether these results are an important addition to the literature. I have some suggestions to improve the paper:

1). The number of subjects expressing inhibitory DREADDs in the chemogenetic experiment is quite low ($n = 5$) and there are no histological images provided to reveal the specificity of the DREADD (or CAV) for the targeted areas. Moreover, there is no functional assessment of the inhibitory effects of CNO on CA1-projecting RE neurons.

2). The 2P imaging experiment is very interesting and runs counter to the expectation one might have about the function of this projection in fear inhibition. By this view, activity in RE terminals might be expected to be highest during the inhibition of fear (when freezing bouts are terminated), but the opposite pattern is observed (and this runs counter to imaging data in RE during remote memory retrieval in mice). The authors suggest that the different time points of memory testing (recent v remote) might account for the results, though there is reason to believe that RE is equally important for the suppression of both recent and remote fear memories. One issue concerns whether terminal fluorescence assessed with 2P actually indexes (or at least correlates with) terminal spiking (or even RE soma spiking). My impression is that it is not clear whether terminal Ca^{++} transients actually reflect underlying spikes. Is it possible that terminal activity assessed with GCaMP is decoupled from RE spiking? Chemogenetic inhibition of RE would presumably reduce terminal activity in CA1 but also increase freezing--so the relationship between terminal activity and freezing behavior would be opposite to what the authors might expect from their terminal imaging dataset.

3) I was expecting the statistical analyses of freezing during the behavioral experiments to involve ANOVAs with within-subject comparison of freezing in the shock and no-shock contexts along with a

between-subject comparisons of VEH or CNO treatment. Instead it appears all the data are measured with pairwise comparisons with nonparametric tests. An ANOVA would allow the authors to determine whether CNO had a differential effect on freezing in shock and no-shock contexts (it does not appear to) using a more appropriate statistical approach.

4) Minor--brain regions should not be capitalized (they are not proper nouns) (lines 49-50). In line 66 "mice fail" should be "animals fail".

Reviewer #2 (Remarks to the Author):

The authors use a clever hM4Di approach, with solid behavioral controls, that helps isolate and selectively inhibit nucleus reuniens (NR) cells that project to dorsal CA1 region of the hippocampus to test the hypothesis that this pathway is involved in the suppression of contextual fear memory retrievals in a novel virtual reality paradigm. In the first set of experiments, the manuscript presents novel data combining these approaches and also importantly replicates similar studies of the role of NR in fear conditioning (esp. generalization, retrieval, and extinction). In these experiments, I have a few questions about the interpretation of the behavioral data as demonstrating a suppression of the retrieval of fear memories and the role of NR-CA1 pathways in extinction circuits (which I don't think was demonstrated). In the second set of experiments, the authors use a very cool in vivo two-photon imaging approach targeting the same NR-CA1 axons that innervate the dorsal CA1 region in the behavioral experiments. They show that these axons ramp up or down activity near behavioral transitions (around freezing behaviors) in a memory-dependent manner but not a content-specific manner. In the third experiment the authors use a computational model to predict NR-CA1 axon activity, which is successful after conditioning but not prior to conditioning. Overall, I believe this manuscript makes an important contribution to understanding the fear memory circuits, esp. the role of NR-CA1 projection pathways.

In general, the text in the figures are too small making the content difficult to follow (especially since the text indicates experimental conditions).

I am confused by the description of the hM4Di agonist approach in the text compared to the depiction in Fig1B. The text says, "This enabled us to intraperitoneally (IP) the hM4Di agonist..." Yet Fig1B shows a cannula in dorsal hippocampus, which I believe is used for axon imaging later. The authors should remove the cannula images from Fig1B as it is confusing when interpreting the behavioral data.

If NR-CA1 inhibited mice freeze more to both contexts, how can their memory retrieval said to be suppressed? On its face this would seem to be enhanced fear retrieval albeit in a non-specific way (like their subsequent NR axonal findings). Importantly here, and highlighted by the authors, is the increase

in freezing lengths is which seem to fit with idea that NR-CA1 is critical for switching between directed memory retrieval modes rather than a suppression of retrieval generally. In this regard NR-CA1 inhibition may be slowing switching between memory strategies prolonging behavior states or causing directed retrievals to be more impoverished and less successful toward generalizing fear. The authors seem to agree with this notion to some degree since they call their findings a suppression of retrieval during ongoing epochs. I suggest rephrasing the title and abstract to get the nuances of these retrieval effects across to the reader better since they are not simple retrieval suppressions.

Lines 203-206. Are the authors claiming that NR-CA1 circuit is necessary part of an extinction learning or retrieval circuit, or that the extinction processes are affected only as a consequence of more freezing on Day 1 in the NR-CA1 inhibition animals?

As a mechanism for extinction, how does contextual fear memory retrieval suppression of NR relate to models whereby fear associations in basolateral amygdala outputs are suppressed before activating central nuclei? Or are these two non-interacting neural circuits processing different aspects of fear extinction?

Is the ramping activity of the NR axons at the beginning and end of freezing transitions consistent with the notion that NR is involved in switching the state of the hippocampus during memory retrievals rather than being a content-specific retrieval mechanism (i.e., context independent)?

The authors primarily discuss the interneuron targets in CA1, however it has also been demonstrated that NR targets principal neurons in CA1. The authors should discuss the role of these targets in relation to increased NR axonal activity and in the context of fear conditioning. Were the authors able to record the activity of any CA1 principal cells and relate their activity to ongoing NR axon activity?

If the authors could further explain the value of the model, it would be helpful because it seems to simply capture the main findings from the axon imaging experiments. Does the model provide a novel insight or predictions?

The authors should comment that only male mice were used much earlier in the manuscript than the experimental methods. For example, it is now common to include sex when reporting any sample sizes such as on Line 109 (N = 20 males).

Reviewer #3 (Remarks to the Author):

This paper presents important and highly significant data indicating the physiological mechanisms of the role in the extinction of contextual fear conditioning for projections from the nucleus reuniens to region CA1 of the hippocampus. The methodology is sound and convincing. The authors present clear data showing that chemogenetic inhibition of the nucleus reuniens inputs reduces the extinction of fear conditioning, and also reduces the ability of animals to show differential freezing between two different virtual contexts (control and shock-associated). They also use sophisticated calcium imaging from axons projecting from nucleus reuniens to CA1 to show that the calcium activity is increased during freezing behavior post-shock, indicating a role of this input in regulating the contextual fear conditioning. This axonal activity is not as well predicted pre-shock. The results of this study are clear and important to understand the important question of how contextual fear conditioning involving the hippocampus is controlled by projections from the nucleus reuniens. I just have some easily addressed comments on the presentation.

Major comments:

1. Page 7 – line 243 – “highly correlated (See Method Details)” - this and other references to “Method Details” should give the specific method subsection heading as it is otherwise very time consuming to find this point in the method details. As noted below, I also think this point needs more elaboration over whether ALL axons are correlated or just a few.
2. Page 24 – line 478 – “correlation” – the point here in the Methods about correlation of axons should be elaborated as the possibility of two axons from the same cell is different from the issue of whether all of the axons in the field of view are correlated. Do they really think that all the axons are correlated?
3. Page 5 – line 156 – If DCZ is not metabolized into clozapine the way that CNO is, then this should be mentioned further here. This is mentioned in the methods but could be mentioned here in the main text as this is an important point that many researchers are ignoring.
4. Page 14 – figure 2E – why does the NR axon activity fall off slightly before the end of freezing? Shouldn't it keep ramping up if it is involved in the termination of freezing? They do discuss this point a bit in the discussion when comparing results with other studies, but in the discussion they might want to at least point out that their activity does ramp up a bit at the start.
5. Page 16 – Figure 3G – what do the graded dot colors indicate? The legend seems to only show two colors (pink and teal) and yet the dots have many colors including orange and light green.

Specific comments:

Page 1 – line 19 – “Contextual fear extinction”- I initially read this as contextual fear conditioning. Might be clearer if you emphasize extinction by saying: “extinction of contextual fear”

Page 2 – line 42 – “therefore” – this seems an odd word here

Page 3 – line 109 – “froze” – Does this include failure to initiate running? This would be a good spot to mention the specific subsection heading that provides the definition of freezing in the methods.

Page 4 – line 133 – “preshocked context” – I did not see the preshock freezing mentioned earlier so this came as a surprise here – They should discuss the freezing in preshock before this point in the text. It is shown in the figure but not mentioned in the text.

Page 10 – line 396 – “since this” – this doesn’t emphasize the contrast. Better to remove the while and say: “However, since this…”

Page 13 – fig 1C – it is odd that the label for the y-axis appears on the top of the graph – would be better to label the y axis with it.

Page 13 – line 482 – “retreival” – spelling ei vs. ie

Page 13 – line 490 – confusing that reference to (D) looks same as the original (D) description. Maybe say “(see part D)”

Page 14 – Fig 2D – Why is activity during running higher in the control context than the shocked in the preshock condition (page 7 line 261)? This is statistically significant but confusing because it should be before the experimental manipulation that differentiates the groups.

Page 21 – line 641, 642 – “In mice…were” – the grammar seems flawed here.

Page 21 – line 667 – They should mention specifically that CNO is metabolized to clozapine which has effects on many receptors (rather than just referring to “off-target effects”)

Page 24 – Line 775 – “averaged” – how much of the smoothness of matching is due to averaging in groups and how much due to averaging over 8000 runs? (minor point)

Reviewer #4 (Remarks to the Author):

The authors examined the effects of nucleus reuniens (NR) at CA1 of the hippocampus on contextual fear memory. Using a virtual reality conditioning paradigm, they importantly showed that chemogenetic suppression of NR projections to CA1 prolonged freezing episodes, induced fear generalization and delayed extinction – thus pointing to a critical role for NR in contextual fear extinction.

This is a generally a well done report although I have reservations about the calcium imaging experiment -- as discussed below.

Comments:

1. They analyzed NR effects on SLM of the dorsal CA1 on contextual fear, but NR projections to dorsal CA1 are approximately 10 fold less pronounced than those to ventral CA1 and the ventral but not the dorsal CA1 projects to the medial prefrontal cortex. Did they examine NR effects on the ventral CA1?

2. They initially discuss two novel environments – one in which mice were shocked and the other not, setting up the two contexts. I am not sure they fully described these two environments?

3. On the DREADD-induced inhibition of NR projections to the dorsal CA1, a very small number of mice were used. For instance, for the NR-CA1 inhibition group, n=5 and for the NR-CA1 intact group, n=8. Further for the NR inhibition group, the 5 mice were divided into shocked and un-shocked groups, leaving only 2 or 3 mice per condition. This is a concern and needs to be addressed.

4. It would important to include some histological sections showing of hM4Di-DREADD injections in NR.

5. Was their statement (line 196) that “even though the NR-CA1 pathway was intact on day 2” meant to indicate that this pathway was not suppressed on the 2nd day? And if so, did they examine the effects of suppressing this pathway on 2nd day for any mice?

6. Regarding the 2 photon Ca⁺⁺ imaging of NR axons at CA1, it is not entirely clear why this was limited to only one axon/mouse for a total of 10 axons/10 mice? Further they state that their analysis was limited to a “putative” single axon per animal? “Putative? Of greatest concern, however, is whether they were truly able to tract/record from the same axons across the 4 days of recording -- without in some

way specifically tagging single axons which was not done. And in this regard in the Methods (lines 699-701) they state, “When possible the same axonal field was returned to across days”. If they are uncertain that the same axonal field was assessed across days, this would seem to invalidate at least some of the results of this experiment. And it would seem, in fact, to be a very difficult task to record from the same axon over 4 days given the variables involved such as precisely positioning the head-plate and microscope from day to day, or movements of the head over 4-day period that could shift (at least slightly so) the position of neurons/axons of the field, etc. Finally, their results showing increases in freezing and decreases with movement following conditioning seems at odds with their demonstration that NR suppression lengthens freezing episodes. And as they discussed, their findings in this regard conflict with those of a recent study examining NR effects on the amygdala in fear conditioning showing that NR stimulation “significantly shortened freezing epochs” (line 369).

7. Related to point 6, they might more fully compare their results to those of Graff and co-workers, focusing on differences (or similarities) of NR effects on the hippocampus vs those on the amygdala in fear/fear extinction.

8. The figures were well done and informative but it is unclear why 5 of the figures were relegated to extended data figures and were not included in the main text of the Results?

We would like to thank the 4 reviewers for their time spent reading and thinking about our manuscript, and the helpful feedback they provided us which we believe has led to significant improvement. Note that the initial submission was tailored as a Brief Communication for Nat. Neuroscience but has now been reformatted to fit better as an Article at Nat. Communications. We made major revisions to the manuscript with new experiments and analysis and expanded it from 3 to 6 main figures. Our responses to reviews are in blue. References cited here are listed at the end of the present document and do not correspond to the reference numbers in the manuscript.

REVIEWER COMMENTS

Reviewer #1 (Remarks to the Author):

This is an exciting paper examining the role for nucleus reuniens projections to hippocampal area CA1 in head-fixed, behaving mice. The authors show in a novel VR task performed on a treadmill that mice exhibit contextual fear conditioning which is manifest in discriminated conditioned freezing in the shock vs no-shock contexts. Chemogenetic inhibition of RE neurons projecting to dorsal hippocampus increased freezing in both the shock and no-shock contexts suggesting this pathway plays a role in inhibiting the expression of hippocampus-dependent fear memory. These effects were not simply due to nonspecific increases in freezing (or decreases in locomotor activity) and are similar to findings previously reported in rats. The authors also used 2P calcium imaging to record from RE terminals in CA1 SLM using axon-targeted GaMP6. These recordings revealed that RE terminal activity became tuned to bouts of freezing after conditioning, exhibiting greater activity during bouts of freezing than locomotion. Ultimately, the data suggest that RE projections to CA1 inhibit hippocampal fear memory and facilitate discrimination and extinction. Altogether these results are an important addition to the literature. I have some suggestions to improve the paper:

Thank you for the positive comments and comprehensive summary of the findings.

1). The number of subjects expressing inhibitory DREADDs in the chemogenetic experiment is quite low ($n = 5$)

Based on this and a similar comment from another reviewer, we have now repeated this experiment in an additional 4 mice, bringing our total to $n = 9$. As you can see in the new Fig. 2D-E and Fig. 3, all the original findings remain unchanged, with the initial differences we reported now strengthened.

and there are no histological images provided to reveal the specificity of the DREADD (or CAV) for the targeted areas.

This is an important comment. We have now added a histological image which is displayed in Fig. 2B showing the specificity of our DREADDs targeted to NR cell bodies and NR axons in CA1 SLM.

Moreover, there is no functional assessment of the inhibitory effects of CNO on CA1-projecting RE neurons.

It is true that we demonstrate a clear behavioral effect of inhibiting NR-CA1 projecting neurons with DCZ, but we do not show a functional decrease in axon activity with DCZ. To do this, the experiment requires dual expression of both Cre-dependent DREADDs and axon-GCaMP in the same cells. Our Cre-dependent DREADDs are selectively expressed in NR neurons projecting to CA1 by injecting a retrograde virus into CA1 in combination with a virus injected into NR that leads to the expression of Cre. To ensure axon-GCaMP is also expressed in the same cells we would need to also inject a virus into NR that leads to the expression of Cre-dependent axon-GCaMP.

However, there is a major issue with this approach. Using 2 viruses that lead to the expression of 2 independent Cre-dependent constructs causes major problems with expression such that neither construct is appropriately expressed in cells. This is a known issue that has been published^{1,2}, where the authors' attempt to obtain co-expression of cre-dependent constructs in the same cells is unsuccessful and they quantify the interference caused by the 2 viruses¹ (see figs 2 and 3) and². It remains unclear why neurons have problems with co-expressing 2 independent cre-dependent constructs from 2 viruses (could be a viral transfection interference problem or a genetic machinery interference problem). Anecdotally, we have tried to do this in the past in other cells and were not successful. Cells can tolerate one virus but not both. We have also talked to other labs that use similar techniques, and they too have had issues with dual expression.

One way around this is to make a new virus that expresses both axon-GCaMP and DREADDs from the same construct. The first issue here is that custom-virus creation takes ~3-5 months, from our experience having custom viruses made in the past. The second issue is that axon-GCaMP takes 10-12 weeks to express in NR-axons in CA1, whereas DREADDs take 3-4 weeks. This makes this approach problematic, as DREADDs will be over-expressed on a 10-12-week timeframe, which can cause major issues with cell health.

Although the reviewer makes a good suggestion, and it would be a nice addition to the paper to observe DCZ-induced inhibition of NR axon activity, it is not currently technically possible in our setup. We think that the clear behavioral effect of DCZ in our increased number of mice (from $n = 5$ to $n = 9$), plus the histology showing selective expression of DREADDs in NR somas and axons in CA1, plus our controls showing that neither the virus carrying DCZ nor DCZ itself impact mouse behavior, is strong evidence that DCZ is selectively inhibiting the NR-CA1 pathway to exert its effects on behavior.

2). The 2P imaging experiment is very interesting and runs counter to the expectation one might have about the function of this projection in fear inhibition. By this view, activity in RE terminals might be expected to be highest during the inhibition of fear (when freezing bouts are terminated), but the opposite pattern is observed (and this runs counter to imaging data in RE during remote memory retrieval in mice). The authors suggest that the different time points of memory testing (recent v remote) might account for the results, though there is reason to believe that RE is equally important for the suppression of both recent and remote fear memories. One issue concerns whether terminal fluorescence assessed with 2P actually indexes (or at least correlates with) terminal spiking (or even RE soma spiking). My impression is that it is not clear whether terminal Ca^{++} transients actually reflect underlying spikes. Is it possible that terminal activity assessed with GCaMP is decoupled from RE spiking?

We thank the reviewer for this interesting idea. It is possible that NR axon activity in CA1 is decoupled from spiking activity in the somas. There is some evidence for this in other circuits and cells (citations provided in Discussion section on this), and we cannot rule it out here. The decoupling between soma and axon would be an interesting investigation (beyond the scope of this paper). It would not, however, change the interpretation of the elevated activity we report here following CFC during freezing epochs. Increased GCaMP activity is interpreted as an increase in release of glutamate from NR terminals, which would have an impact on post-synaptic partners of NR-axons in CA1, and this holds true whether the soma is also spiking along with the axons. Based on this interesting idea put forth by the reviewer, we have added the following comment to our discussion section:

"Another possibility for divergent results is that our axonal measurements are recording localized axonal spiking activity decoupled from somatic spiking activity, which has been observed in other circuits^{59,60}. In this case, local synaptic modulation in CA1 may be exerting excitatory or inhibitory impacts on NR-axon activity, decoupling its activity profile in CA1 from somatic activity in NR."

Chemogenetic inhibition of RE would presumably reduce terminal activity in CA1 but also increase freezing--so the relationship between terminal activity and freezing behavior would be opposite to what the authors might expect from their terminal imaging dataset.

The reviewer is correct that without the axon activity data we might have expected the opposite of what we found. However, we propose a framework in the paper that reconciles these ideas. Our framework is that when mice freeze following CFC, NR axon terminals increase their activity, and we believe this activity is somehow disrupting the process of fear memory retrieval in CA1. In a sense, the NR terminals are acting as a competing signal that acts to interfere with memory retrieval dynamics, biasing the animal to stop retrieving the fear memory and therefore promoting movement. Our DREADD inhibition experiment is

reducing the activity of NR axon terminals in CA1 during freezing epochs following CFC. In this experiment we are taking away the disruptive NR signal, which allows the CA1 to continue to retrieve the fear memory and thus prolongs freezing epochs. Another way to think about this is that there are two competing forces – fear memory retrieval and fear memory suppression. Both forces can occur at the same time and are in competition. During freezing following CFC the retrieval force initially wins, causing fear, but the opposing suppression force turns on and is pushing back to suppress the memory. At some point the suppression force wins and the animal starts moving. During the DREADD inhibition experiment we are effectively reducing the power of the suppression force allowing the memory retrieval force to exert its effects for longer.

Based on the reviewer's comment and similar comments from another reviewer, we have added a new supplementary figure that brings all our findings together in one place depicted as cartoon model to help visualize our findings and framework. The figure and the caption together describe our framework for how NR-CA1 suppresses ongoing fear memory retrieval that also leads to fear extinction and context discrimination (see Supplementary Fig. 5).

3) I was expecting the statistical analyses of freezing during the behavioral experiments to involve ANOVAs with within-subject comparison of freezing in the shock and no-shock contexts along with a between-subject comparisons of VEH or CNO treatment. Instead it appears all the data are measured with pairwise comparisons with nonparametric tests. An ANOVA would allow the authors to determine whether CNO had a differential effect on freezing in shock and no-shock contexts (it does not appear to) using a more appropriate statistical approach.

We thank the reviewer for this comment. Based on this we have now re-analyzed all our behavioral data using a three-way repeated measures ANOVA with *post-hoc* testing on relevant pairwise comparisons. The Methods: Statistics section has been appropriately updated. While specific values and degrees of significance have been changed, no major findings from the paper have changed as a result of this updated statistical methodology, giving us increased confidence in our overall findings.

4) Minor--brain regions should not be capitalized (they are not proper nouns) (lines 49-50). In line 66 "mice fail" should be "animals fail".

Corrected.

Reviewer #2 (Remarks to the Author):

The authors use a clever hM4Di approach, with solid behavioral controls, that helps isolate and selectively inhibit nucleus reuniens (NR) cells that project to dorsal CA1 region of the

hippocampus to test the hypothesis that this pathway is involved in the suppression of contextual fear memory retrievals in a novel virtual reality paradigm. In the first set of experiments, the manuscript presents novel data combining these approaches and also importantly replicates similar studies of the role of NR in fear conditioning (esp. generalization, retrieval, and extinction). In these experiments, I have a few questions about the interpretation of the behavioral data as demonstrating a suppression of the retrieval of fear memories and the role of NR-CA1 pathways in extinction circuits (which I don't think was demonstrated). In the second set of experiments, the authors use a very cool in vivo two-photon imaging approach targeting the same NR-CA1 axons that innervate the dorsal CA1 region in the behavioral experiments. They show that these axons ramp up or down activity near behavioral transitions (around freezing behaviors) in a memory-dependent manner but not a content-specific manner. In the third experiment the authors use a computational model to predict NR-CA1 axon activity, which is successful after conditioning but not prior to conditioning. Overall, I believe this manuscript makes an important contribution to understanding the fear memory circuits, esp. the role of NR-CA1 projection pathways.

We thank the reviewer for their detailed insights and comments on our paper.

In general, the text in the figures are too small making the content difficult to follow (especially since the text indicates experimental conditions).

Figure text size has been increased wherever possible.

I am confused by the description of the hM4Di agonist approach in the text compared to the depiction in Fig1B. The text says, "This enabled us to intraperitoneally (IP) the hM4Di agonist..." Yet Fig1B shows a cannula in dorsal hippocampus, which I believe is used for axon imaging later. The authors should remove the cannula images from Fig1B as it is confusing when interpreting the behavioral data.

This is a good observation, and we agree it was confusing. We have now associated the cannula implantation image with the data regarding axon imaging in the new main figures. This clarification has been reflected in the figure text for Fig. 2A and 4A.

If NR-CA1 inhibited mice freeze more to both contexts, how can their memory retrieval said to be suppressed?

Thank you for the opportunity to clarify. We are claiming that NR-CA1 suppresses fear memory retrieval under **normal** conditions, i.e. when it is **not** inhibited by DREADDs. We show that when we inhibit NR-CA1 we reduce fear memory suppression, thus promoting fear memory retrieval as demonstrated by longer freezing epochs. This supports NR-CA1's role in normally suppressing fear memory retrieval.

We have now added text to help clarify this in the results section discussing the differences between NR-inhibited and NR-intact mice:

“Our results overall indicate that the intact NR-CA1 pathway sends a potent fear suppression signal, critical for shortening the length of freezing epochs, preventing fear generalization, and inducing rapid extinction of contextual fear. Inhibiting this fear suppression signal, therefore, induces a higher fear response by increasing the length of freezing epochs, increasing fear generalization, and decreasing contextual fear extinction.”

We have also added a new supplementary figure that brings all our findings together in one place depicted as cartoon model to help visualize our findings and framework. The figure and the caption together describe our framework for how NR-CA1 suppresses ongoing fear memory retrieval that also leads to fear extinction and context discrimination (see Supplementary Fig. 5).

On its face this would seem to be enhanced fear retrieval albeit in a non-specific way (like their subsequent NR axonal findings).

Yes, exactly. Inhibiting NR-CA1 does indeed enhance fear memory retrieval as we are removing the fear suppression function of the NR-CA1 pathway. This induces increased freezing in both the appropriate (shocked) and inappropriate (control) contexts.

Importantly here, and highlighted by the authors, is the increase in freezing lengths is which seem to fit with idea that NR-CA1 is critical for switching between directed memory retrieval modes rather than a suppression of retrieval generally.

We agree with the reviewer. The NR-CA1 under normal conditions (when not DREADD-inhibited) suppresses memory retrieval in inappropriate contexts (in this case the “control” context). This allows for context discrimination (memory specificity), which we show is lost under DREADD inhibition. However, NR-CA1 is also suppressing fear memory retrieval to some extent in the feared environment under normal conditions. This is shown by the result that under DREADD inhibition, freezing lengths are increased. Our framework is that NR-CA1 is always providing a fear memory suppression force in both appropriate and inappropriate contexts. The fear memory retrieval force is just much greater in the feared context than in the control context, so it can completely suppress fear in the control context but it competes with a strong fear memory retrieval force in the feared context, so can only reduce fear, not eliminate fear.

We now point to text in the discussion that clarifies this idea:

"The function of the NR-CA1 pathway in CFMR suppression is not restricted to the context in which shocks were presented but extends to similar contexts where shocks never occurred. This seems to limit overgeneralization as shown by NR-CA1 inhibition reducing context discrimination. Lastly, the process of suppressing ongoing CFMR by the NR-CA1 pathway also has longer term effects, as shown by prolonged fearful behavior and delayed extinction following NR-CA1 inhibition. In summary, our observations support a framework in which the NR-CA1 pathway actively suppresses fear responses by disrupting ongoing hippocampal-dependent CFMR to promote non-fearful behavior, and this process also limits overgeneralization and promotes fear extinction."

In this regard NR-CA1 inhibition may be slowing switching between memory strategies prolonging behavior states or causing directed retrievals to be more impoverished and less successful toward generalizing fear. The authors seem to agree with this notion to some degree since they call their findings a suppression of retrieval during ongoing epochs. I suggest rephrasing the title and abstract to get the nuances of these retrieval effects across to the reader better since they are not simple retrieval suppressions.

We do agree that NR-CA1 inhibition is slowing switching of memory retrieval states such that mice get "stuck" in fear memory retrieval for longer without NR-CA1 to push the network into a different state or suppress/disrupt the specific dynamics associated with fear memory retrieval. However, exactly how the NR-CA1 interacts with CA1 dynamics is not investigated in this paper, only discussed in the Discussion section. We use the term suppression in the title and abstract and throughout the paper because behaviorally the data supports the idea that the NR-CA1 pathway suppresses fear responses. This is also the term used by others describing the role of the NR itself (not the NR-CA1 pathway). To connect our work to the rest of the literature on NR function, we chose to use the term suppression. We are working on investigating the specific CA1 population dynamics that are influenced by the NR-CA1 pathway, and this will provide insight into how NR changes hippocampal processing. But without direct evidence for memory switching or any other hippocampal dynamics, we felt it best to use the term suppression as it fits with the current known function of the NR itself.

The reviewer is right that the title does not capture all the nuances of the main findings of the paper. We have therefore changed the title to incorporate the findings that the NR-CA1 not only suppresses and ongoing fear memory retrieval event, but also promotes extinction and context discrimination. The new title is now:

A Thalamic-Hippocampal CA1 Signal for Contextual Fear Memory Suppression, Extinction, and discrimination

Lines 203-206. Are the authors claiming that NR-CA1 circuit is necessary part of an

extinction learning or retrieval circuit, or that the extinction processes are affected only as a consequence of more freezing on Day 1 in the NR-CA1 inhibition animals?

This is an interesting consideration, and we thank the reviewer for bringing it up. The reviewer is pointing out that when NR-CA1 is inhibited on day 1 it leads to much higher freezing behavior on day 1 and on day 2 there is reduced extinction. This could be an indirect effect of the increased freezing behavior on day 1 or it could be due to a disruption in neural processes on day 1 that are necessary for normal levels of extinction on day 2. Our data does not allow for these two hypotheses to be teased apart. Regardless of which mechanism is at play, it does not change the conclusion that the NR-CA1 pathway is necessary for normal extinction, which is strongly supported by our data. We have now clarified these ideas in the discussion.

As a mechanism for extinction, how does contextual fear memory retrieval suppression of NR relate to models whereby fear associations in basolateral amygdala outputs are suppressed before activating central nuclei? Or are these two non-interacting neural circuits processing different aspects of fear extinction?

We do think that inevitably the suppression of fear memory in CA1 changes its interaction with the amygdala or other brain regions connected to amygdala. Our current working hypothesis is when CA1 is retrieving fear memories it has an output, either directly or indirectly, to Amygdala that becomes associated with fear circuits there (maybe in BLA). When NR suppresses CA1, the CA1 output is altered and therefore does not activate amygdala fear circuits. In this framework, the BLA-central nuclei model of fear suppression is not necessarily part of the process but is instead an independent circuit of memory suppression that may be involved in non-hippocampal-dependent memory suppression, i.e., non-contextual fear memories such as those associated with a tone. The specific circuit mechanisms that relate hippocampal fear memories with amygdala are very interesting. We have now added a comment on this topic in the discussion.

Is the ramping activity of the NR axons at the beginning and end of freezing transitions consistent with the notion that NR is involved in switching the state of the hippocampus during memory retrievals rather than being a content-specific retrieval mechanism (i.e., context independent)?

We think it is involved in switching the state of the CA1 to suppress fear memory retrieval as it only starts to ramp up after the animal is in a state of fear memory retrieval, i.e., it doesn't cause the memory retrieval it responds to it. The ramping up of the NR-CA1 signal after animals start a freezing epoch is consistent with its role in suppressing the fear memory once it starts (see new Supplementary Fig. 5 and caption for details). The NR-CA1 axon signals we recorded during freezing epochs is consistent with NR-induced memory suppression, as described above in our framework regarding NR providing a suppression

force in CA1 that competes with the fear memory retrieval force and described in our new Supplementary Fig. 5. In this framework, it seems to make sense that when mice first enter a fear memory retrieval state that the NR-CA1 suppression signal is initiated and begins to ramp up. This signal remains high until the CA1 begins to transition to a non-fear memory retrieval state (possibly by disrupting replay events as discussed in our Discussion section). At this point the activity in the NR-CA1 pathway starts to reduce, possibly through a feedback pathway from CA1 to NR, which anatomically exists. For instance, as fear memory retrieval starts to decay towards the end of a freezing epoch, the input from CA1 to NR also reduces, which could result in reduced NR-CA1 activity right before the animal starts to move. The feedback pathway from CA1-NR remains to be investigated, but experiments on this pathway would help test the validity of our current framework. So the ramping up and down of the NR-CA1 signal during freezing is consistent with its role as a fear memory suppression signal. Further, if the NR-CA1 pathway were instead a content-specific fear memory retrieval mechanism, inhibiting it would reduce retrieval, but it does the opposite and prolongs fear memory retrieval (as shown by longer freezing epochs). We have elaborated on this in places throughout the text and added a new supplementary figure (5) that we hope helps make our ideas clear.

The authors primarily discuss the interneuron targets in CA1, however it has also been demonstrated that NR targets principal neurons in CA1. The authors should discuss the role of these targets in relation to increased NR axonal activity and in the context of fear conditioning.

The reviewer is correct that NR likely interacts with both interneurons and directly with pyramidal cells in CA1. However, since there is active controversy in the field over if NR axons directly synapse onto CA1 pyramidal dendrites, we wanted to include a discussion of these interneurons which are well-established post-synaptic partners of NR-axons in CA1. We have also discussed how NR could exert its effects on the CA1 circuit through direct contact onto excitatory apical tuft dendrites of CA1 pyramidal cells in our discussion section. We have now modified language to ensure it is clear which synaptic type we are referring to.

Were the authors able to record the activity of any CA1 principal cells and relate their activity to ongoing NR axon activity?

This is an excellent idea and something we are particularly interested in. Indeed, recordings of CA1 pyramidal cells and dendrites during CFC and retrieval with and without NR inhibition are already underway. However, these experiments are complex and require careful controls and detailed population level analysis of complex dynamics. We believe this is beyond the scope of this paper and will be best explored in detail in a follow-up paper.

If the authors could further explain the value of the model, it would be helpful because it

seems to simply capture the main findings from the axon imaging experiments. Does the model provide a novel insight or predictions?

We thank the reviewer for bringing this up. While by eye it seemed clear to us that the NR-CA1 pathway had preferential activity for freezing epochs, it was also possible that this activity was caused by other related factors, such as pupil diameter or interactions of other behavioral dynamics. Pupil dynamics were of specific interest as other thalamic regions have been shown to encode for attention, and we wanted to rule out changes in pupil diameter during freezing, potentially induced by higher attention during freezing epochs, being correlated to our NR-axon signal. Our model enables us to make an unbiased assessment of which behavioral factors best predicted NR-axon activity, which we believe greatly strengthens our argument that NR-CA1 axons selectively tune to freezing post-shocks. Additionally, the model provides detailed information on the timing of NR-axonal tuning, showing that future velocity shifted forward by 0.5 s was most-commonly used to predict NR activity, information we did not have before running our model. Our reasoning for using this modeling approach has been added to the paper in the section 'Encoding Model Predicts NR-CA1 Axonal Activity, But Only Following CFC' in the first paragraph.

The authors should comment that only male mice were used much earlier in the manuscript than the experimental methods. For example, it is now common to include sex when reporting any sample sizes such as on Line 109 (N = 20 males).

We thank the reviewer for pointing this out. We now refer to the male mice in the abstract and throughout the text.

Reviewer #3 (Remarks to the Author):

This paper presents important and highly significant data indicating the physiological mechanisms of the role in the extinction of contextual fear conditioning for projections from the nucleus reuniens to region CA1 of the hippocampus. The methodology is sound and convincing. The authors present clear data showing that chemogenetic inhibition of the nucleus reuniens inputs reduces the extinction of fear conditioning, and also reduces the ability of animals to show differential freezing between two different virtual contexts (control and shock-associated). They also use sophisticated calcium imaging from axons projecting from nucleus reuniens to CA1 to show that the calcium activity is increased during freezing behavior post-shock, indicating a role of this input in regulating the contextual fear conditioning. This axonal activity is not as well predicted pre-shock. The results of this study are clear and important to understand the important question of how contextual fear conditioning involving the hippocampus is controlled by projections from the nucleus reuniens. I just have some easily addressed comments on the presentation.

We thank the reviewer for their positive perspective on our paper.

Major comments:

1. Page 7 – line 243 – “highly correlated (See Method Details)” - this and other references to “Method Details” should give the specific method subsection heading as it is otherwise very time consuming to find this point in the method details.

We have now addressed this and any references to Methods now have subsection titles throughout the paper.

As noted below, I also think this point needs more elaboration over whether ALL axons are correlated or just a few.

NR-axons in CA1 run parallel to our cannula window, and branch significantly within CA1. Our FOVs are relatively small (under 200x200um) so that we can record a high enough $\Delta f/f$ from small axonal structures 1-2 μm wide. Therefore, each FOV only has one branching axon that is reliably active – meaning it has a significant signal above noise (above 10% $\Delta f/f$). There may be other axons in the FOV that are not active above this level. To clarify this point, we added Fig. 5A which demonstrates the ROIs used to make up a single axonal signal, illustrating a qualitative example of the above points. We chose to take a conservative approach to single axon identification by setting a high correlation threshold between axon ROIs – meaning axon segment activity must be highly correlated to be considered part of the same axon. Using our approach, each FOV typically contained one ensemble of highly correlated axon ROIs which we consider to be from a single axon (on some occasions, like the example shown in Fig. 5A, you can see a single branching axon). We expanded discussion of this decision in the Methods section: Image Processing and ROI Selection.

2. Page 24 – line 478 – “correlation” – the point here in the Methods about correlation of axons should be elaborated as the possibility of two axons from the same cell is different from the issue of whether all of the axons in the field of view are correlated. Do they really think that all the axons are correlated?

As described in our above comment, we take a conservative approach to axon identification in an attempt to resolve this issue. By setting a high correlation threshold and limiting our axon analysis to a single putative axon per FOV, comprised largely of a single large branching axon with connected ROI segments identified in Suite2p (and visually conformed by eye), we believe we have identified and analyzed a single branching axon from a single NR neuron in each FOV.

3. Page 5 – line 156 – If DCZ is not metabolized into clozapine the way that CNO is, then this should be mentioned further here. This is mentioned in the methods but could be

mentioned here in the main text as this is an important point that many researchers are ignoring.

This is a good point. We have elaborated on this in the main text in the first paragraph of our NR-inhibition experiment results section.

4. Page 14 – figure 2E – why does the NR axon activity fall off slightly before the end of freezing? Shouldn't it keep ramping up if it is involved in the termination of freezing?

This is a good question and deserves some thought and discussion. We think the drop in the signal at the end of freezing epochs could indicate a neural preparation or decision-making phase during which fear memory retrieval in the hippocampus has been suppressed but the decision to move has yet to occur. It also takes time for the brain to prime the motor circuits and coordinate the necessary movements before the actual initiation of movement. The CA1 is not a motor region, so the suppression of a fear memory in this region is unlikely to immediately cause movement. The decrease in the ramping signal may reflect the completion of fear memory suppression, allowing the animal to respond quickly when the decision to move is made, but the decision may take time. Animals may engage in an internal evaluation or computation of relevant information, such as assessing potential risks or benefits associated with the impending movement. The decision-making processes and the subsequent activation of motor commands may be occurring as the NR-CA1 signal is ramping down. We have now added a full paragraph to the Discussion section describing this idea.

They do discuss this point a bit in the discussion when comparing results with other studies, but in the discussion they might want to at least point out that their activity does ramp up a bit at the start.

We agree that this is an important observation. We have now elaborated on this in the Discussion section by adding more detail and expanding what it means.

5. Page 16 – Figure 3G – what do the graded dot colors indicate? The legend seems to only show two colors (pink and teal) and yet the dots have many colors including orange and light green.

We apologize for the confusion. The graded dots are for each individual mouse run through the model. This has been elaborated upon in the figure legend to make it clearer.

Specific comments:

Page 1 – line 19 – “Contextual fear extinction”- I initially read this as contextual fear conditioning. Might be clearer if you emphasize extinction by saying: “extinction of contextual fear”

We agree with the reviewer and have therefore changed all references to this to “extinction of contextual fear”.

Page 2 – line 42 – “therefore” – this seems an odd word here

Changed.

Page 3 – line 109 – “froze” – Does this include failure to initiate running? This would be a good spot to mention the specific subsection heading that provides the definition of freezing in the methods.

Changed.

Page 4 – line 133 – “preshocked context” – I did not see the preshock freezing mentioned earlier so this came as a surprise here – They should discuss the freezing in preshock before this point in the text. It is shown in the figure but not mentioned in the text.

We now mention earlier in the manuscript that animals stop running and become immobile sometimes in the preshock condition in the results section. Additionally, we have decided to show our baseline-normalized figures for freezing quantities as the main behavioral figures in the paper, instead of in the supplementary figures. We believe this highlights the behavioral findings of the paper, which are an increase in freezing epochs due to shocks as compared to pre-shock baseline freezing quantities with and without NR-CA1 intact.

Page 10 – line 396 – “since this” – this doesn’t emphasize the contrast. Better to remove the while and say: “However, since this...”

Good point, this has now been changed.

Page 13 – fig 1C – it is odd that the label for the y-axis appears on the top of the graph – would be better to label the y axis with it.

We thank the reviewer for pointing this out and agree that it was confusing. ‘Mouse position on the virtual track’ was the title for the subfigure, however, we agree that it is confusing to a reader. We have changed this to ‘example mouse behavior’ throughout the figures where such data are presented. The y axis is labeled with 0m and 2m at the top and bottom, and the x axis with time in seconds.

Page 13 – line 482 – “retrieval” – spelling ei vs. ie

Thanks for pointing this out. We have made sure it is now spelled correctly throughout the text.

Page 13 – line 490 – confusing that reference to (D) looks same as the original (D) description. Maybe say “(see part D)”

Changed.

Page 14 – Fig 2D – Why is activity during running higher in the control context than the shocked in the preshock condition (page 7 line 261)? This is statistically significant but confusing because it should be before the experimental manipulation that differentiates the groups.

This is an interesting observation. Indeed, the activity during running should be the same in both contexts before CFC during running, as they are during immobility. The reviewer has noticed a slightly higher level of activity during running pre-CFC in the control context versus the shocked context pre-shocks. It is unclear why this is the case. Importantly, the slightly higher activity during running pre-CFC is opposite to our main observation following CFC that NR axon activity becomes highly tuned to freezing epochs and therefore comparatively reduced during running, so this does not affect our main conclusion. Further, while we do observe a slight, but statistically significant difference when grouping all 10 axons, we do not observe this difference in the subset of 4 axons we tracked across days (Fig. 5D), indicating that this may simply be due to the limited number of axons we recorded from ($n = 10$; a consequence of the technically challenging nature of these experiments).

Page 21 – line 641, 642 – “In mice...were” – the grammar seems flawed here.

Corrected.

Page 21 – line 667 – They should mention specifically that CNO is metabolized to clozapine which has effects on many receptors (rather than just referring to “off-target effects”)

Added.

Page 24 – Line 775 – “averaged” – how much of the smoothness of matching is due to averaging in groups and how much due to averaging over 8000 runs? (minor point)

We show each r^2 value for each run in Fig. 6B to be transparent about individual run values. All averages are made within-mouse first, then averaged across mice, to demonstrate the most typical run r^2 . Despite the same run quantities being used for the pre-shock day, the r^2 is significantly lower and near-zero for this day, indicating that increased performance in subsequent days is not due to averaging across many runs.

Reviewer #4 (Remarks to the Author):

The authors examined the effects of nucleus reuniens (NR) at CA1 of the hippocampus on contextual fear memory. Using a virtual reality conditioning paradigm, they importantly showed that chemogenetic suppression of NR projections to CA1 prolonged freezing episodes, induced fear generalization and delayed extinction – thus pointing to a critical role for NR in contextual fear extinction.

This is a generally a well done report although I have reservations about the calcium imaging experiment -- as discussed below.

We thank the reviewer for their in-depth and insightful comments.

Comments:

1. They analyzed NR effects on SLM of the dorsal CA1 on contextual fear, but NR projections to dorsal CA1 are approximately 10 fold less pronounced than those to ventral CA1 and the ventral but not the dorsal CA1 projects to the medial prefrontal cortex. Did they examine NR effects on the ventral CA1?

This is a very good point, and investigating the NR-vCA1 would be an exciting project. Although we are aware of the higher density of NR projections to ventral CA1 (vCA1), however, imaging NR axons in vCA1 with 2-photon microscopy is currently not possible due to the requirement of a GRIN lens to access vCA1, which is much deeper in the brain compared to dCA1³. GRIN lenses lower imaging resolution below the level needed to resolve individual axons due the following reasons. They introduce optical aberrations and increased light scattering. A mismatch in the refractive indices of the GRIN lens and brain tissue can also lead to additional image distortions and scattering, further reducing image resolution, and the numerical aperture (NA) of a GRIN lens is lower compared to conventional high-NA objective lenses used in 2-photon microscopy. There is the additional problem of inserting the GRIN lens deep into the brain to access vCA1. Due to pathway of axons projecting from NR to vCA1, the GRIN lens would likely damage NR axons coursing through iCA1 SLM towards vCA1.

While these challenges do not render two-photon imaging completely impossible for imaging the ventral hippocampus, they do make it more technically demanding and may require specialized approaches, imaging systems, or advanced image correction algorithms to overcome the limitations and obtain meaningful data. We are attempting to develop methods that may allow us to investigate the ventral hippocampus as examining the role of NR in vCA1 is of great interest to us, but it will take a considerable amount of time and may not work. This is partly why we have focused this study on the dorsal CA1. Because we have

a strong behavioral result implicating the NR-dCA1 pathway in CFC memory retrieval and extinction, we think the underlying circuitry is worth investigating and reporting. It will be interesting to determine in the future whether the NR-vCA1 pathway has a similar or distinct function, when technology permits such an investigation.

2. They initially discuss two novel environments – one in which mice were shocked and the other not, setting up the two contexts. I am not sure they fully described these two environments?

We thank the reviewer for pointing this out. VR contexts have now been described in much more detail in the Methods: Behavior section, and the code creating the VR contexts have been added to github repo for reproducibility:

<https://github.com/hmacomber/NR-Analysis>.

3. On the DREADD-induced inhibition of NR projections to the dorsal CA1, a very small number of mice were used. For instance, for the NR-CA1 inhibition group, n=5 and for the NR-CA1 intact group, n=8. Further for the NR inhibition group, the 5 mice were divided into shocked and un-shocked groups, leaving only 2 or 3 mice per condition. This is a concern and needs to be addressed.

Based on this reviewer's concern, which was shared by reviewer 1, the NR inhibition group has now been increased from n=5 to n=9. As we discussed in response to reviewer 1, all the original findings remain unchanged with the addition of the new data (see new Fig. 2D-E and Fig. 3), and the initial differences we reported are now strengthened. In response to the concern that the NR-inhibition group were divided into shocked and unshocked groups, we thank the reviewer for the opportunity to clarify this. The n=5 NR-inhibited group was not subdivided to an n=2 or n=3 sub-group. The 5 mice were all in the shocked group, and a separate n=4 mice were used as a non-shocked control group. In the revised version of the manuscript, we have added 4 new mice to this non-shocked group, bringing it to n=8. We have now clarified this in the Methods, figure captions, and results sections.

4. It would be important to include some histological sections showing of hM4Di-DREADD injections in NR.

This is an important comment. We have now added a histological image which is displayed in Fig. 2B showing the specificity of our DREADDs targeted to NR cell bodies and NR axons restricted to CA1 SLM.

5. Was their statement (line 196) that "even though the NR-CA1 pathway was intact on day 2" meant to indicate that this pathway was not suppressed on the 2nd day?

Yes, this is correct. NR-CA1 inhibition only occurred on day 1. We have now clarified this point in the main text.

And if so, did they examine the effects of suppressing this pathway on 2nd day for any mice?

No, we did not. We agree with the reviewer that this would be an interesting experiment. However, adding this experiment would require a full set of new inhibitions and controls, which we think is beyond the scope of this paper. This question could be addressed in future experiments, as it is a worthwhile endeavor that could lead to important insights.

6. Regarding the 2 photon Ca⁺⁺ imaging of NR axons at CA1, it is not entirely clear why this was limited to only one axon/mouse for a total of 10 axons/10 mice?

We thank the reviewer for pointing this out, giving us a chance to clarify. We limited to one axon per mouse for several reasons explained in the methods, which we will also clarify here. NR-axons in CA1 run parallel to our cannula window, and branch significantly within CA1. Our FOVs are relatively small (under 200x200um) so that we can record a high enough $\Delta f/f$ from small axonal structures 1-2 μm wide. Therefore, each FOV only has one branching axon that is reliably active – meaning it has a significant signal above noise (above 10% $\Delta f/f$). There may be other axons in the FOV that are not active above this level. To clarify this point, we added Fig. 5A which demonstrates the ROIs used to make up a single axonal signal, illustrating a qualitative example of the above points. We chose to take a conservative approach to single axon identification by setting a high correlation threshold between axon ROIs – meaning axon segment activity must be highly correlated to be considered part of the same axon. Using our approach, each FOV typically contained one ensemble of highly correlated axon ROIs which we consider to be from a single axon (on some occasions, like the example shown in Fig. 5A, you can see a single branching axon). We expanded discussion of this decision in the Methods section: Image Processing and ROI Selection.

Further they state that their analysis was limited to a “putative” single axon per animal?
“Putative?”

The word putative has been removed for clarity. We additionally were conservative in our across-day tracking, which is in Figure 5. Additional information on cross-day tracking identification has been added to the Methods.

Of greatest concern, however, is whether they were truly able to track/record from the same axons across the 4 days of recording -- without in some way specifically tagging single axons which was not done. And in this regard in the Methods (lines 699-701) they state, “When possible the same axonal field was returned to across days”. If they are uncertain that the same axonal field was assessed across days, this would seem to invalidate at least some

of the results of this experiment. And it would seem, in fact, to be a very difficult task to record from the same axon over 4 days given the variables involved such as precisely positioning the head-plate and microscope from day to day, or movements of the head over 4-day period that could shift (at least slightly so) the position of neurons/axons of the field, etc.

This is an important point and we thank the reviewer for giving us an opportunity to clarify. It is indeed difficult to track the same axons across days. Hence, we were only able to do this in a subset of mice (4 out of 10). We first did our across-days analysis on NR axons without making any assumptions or claims that they were the same axons across days. This analysis is the new Fig. 4. We then did the same analysis on a subset of these axons (n=4) that we do claim are the same axons across days. This analysis is shown in the new Fig. 5 and matches the Fig. 4 results. Tracking either the same axons across days or recording from random NR axons on each day gives the same result, that NR axons become tuned to freezing epochs following CFC.

To track the same axons across days, we create a template at the end of the first imaging day to align subsequent days to by average across frames to get a high SNR image. Each day the template is used to find the same FOV containing the same axon segments. Headplates are chronically attached to the animal's skull, so do not move from day-to-day. The headplate attachment apparatus used to secure animals above the treadmill also do not move other than on a pivot that swings in and out of a fixed position on the right-hand side, so that mice can be placed on and off the treadmill. This means the animal's head is put in the same position each day at the same angle relative to the microscope objective. We then zero the microscope objective relative to the tissue surface and then move down in the z-plane the same distance as the previous day. This procedure gets us close to the targeted FOV. We then perform slight adjustments with our template FOV displayed adjacent to our real-time imaging FOV to match them up. Other than the axon segments in the template FOV, other structures are present that aid with alignment such as blood vessels and bright autofluorescence structures that do not change in fluorescence day-to-day. These structures are very useful for obtaining a precise alignment of the FOV. Lastly, our microscope objective can rotate through a high range of angles. This is not usually required, but if needed we can adjust the angle of our objective across days to keep the imaging plane aligned in z. Fig. 5 shows an example axon recorded over 4 days. You can clearly see it is the same axon as it has the same branching structure. We routinely track the same populations of neurons across days (see⁴), and the same dendrites (see⁵). Axons are more difficult, but our approach allows us to do this with a high rate of success.

Finally, their results showing increases in freezing and decreases with movement [we presume the reviewer is referring to axon activity here and not behavior] following conditioning seems at odds with their demonstration that NR suppression lengthens freezing episodes.

We appreciate the reviewer's perspective on this but do not think these two findings are necessarily at odds with each other. We propose a framework in the paper that reconciles these ideas (which is also described in answer to a comment from reviewer 1). Our framework is that when mice freeze following CFC, NR axon terminals increase their activity, and we believe this activity is somehow disrupting the process of fear memory retrieval in CA1. In a sense, the NR terminals are acting as a competing signal that acts to interfere with memory retrieval dynamics, biasing the animal to stop retrieving the fear memory and therefore promoting movement. Our DREADD inhibition experiment is reducing the activity of NR axon terminals in CA1 during freezing epochs following CFC. In this experiment we are taking away the disruptive NR signal, which allows the CA1 to continue to retrieve the fear memory and thus prolong freezing epochs. Another way to think about this is that there are two competing forces – fear memory retrieval and fear memory suppression. Both forces can occur at the same time and are in competition. During freezing following CFC, the retrieval force initially wins, causing fear, but the opposing suppression force turns on and is pushing back to suppress the memory. At some point the suppression force wins and the animal starts moving. During the DREADD inhibition experiment we are effectively reducing the power of the suppression force allowing the memory retrieval force to exert its effects for longer.

We have now added a new supplementary figure 5 that helps visualize this framework as well as describe our ideas in the figure caption.

And as they discussed, their findings in this regard conflict with those of a recent study examining NR effects on the amygdala in fear conditioning showing that NR stimulation "significantly shortened freezing epochs" (line 369).

We thank the reviewer for bringing this up. However, we do not think our results conflict in behavioral outcome with the Graff et al. study. We used DREADDS to **inhibit** NR, which increased freezing epochs. They used optogenetics to **activate** NR, which decreased freezing epochs. Since we modified NR in opposing directions, these outcomes align with one another. Other whole NR inactivation aligns with inactivation causing more freezing and preventing extinction^{6,7}. We added more to the discussion comparing our study with Graff et al. to clarify our position on this.

7. Related to point 6, they might more fully compare their results to those of Graff and co-workers, focusing on differences (or similarities) of NR effects on the hippocampus vs those on the amygdala in fear/fear extinction.

The reviewer is right that the Graff et al. study is the most relevant study to ours, so it is worthy of more discussion than we had originally provided. We have now expanded our discussion on this paper significantly.

8. The figures were well done and informative but it is unclear why 5 of the figures were relegated to extended data figures and were not included in the main text of the Results?

We thank the reviewer for pointing this out. The initial submission was formatted for Nature Neuroscience as a Brief Communication. Being considered now as an article for Nature Communications has allowed us to expand the main figures. Figures have now been reformatted to 6 main figs and 4 supplemental figs, which we believe makes for a much clearer presentation of the data.

Rebuttal References:

- 1 Saunders, A., Johnson, C. A. & Sabatini, B. L. Novel recombinant adeno-associated viruses for Cre activated and inactivated transgene expression in neurons. *Front Neural Circuits* **6**, 47, doi:10.3389/fncir.2012.00047 (2012).
- 2 Saunders, A. & Sabatini, B. L. Cre Activated and Inactivated Recombinant Adeno-Associated Viral Vectors for Neuronal Anatomical Tracing or Activity Manipulation. *Curr Protoc Neurosci* **72**, 1 24 21-21 24 15, doi:10.1002/0471142301.ns0124s72 (2015).
- 3 Biane, J. S. *et al.* Neural dynamics underlying associative learning in the dorsal and ventral hippocampus. *Nat Neurosci* **26**, 798-809, doi:10.1038/s41593-023-01296-6 (2023).
- 4 Dong, C., Madar, A. D. & Sheffield, M. E. J. Distinct place cell dynamics in CA1 and CA3 encode experience in new environments. *Nat Commun* **12**, 2977, doi:10.1038/s41467-021-23260-3 (2021).
- 5 Sheffield, M. E. & Dombeck, D. A. Calcium transient prevalence across the dendritic arbour predicts place field properties. *Nature* **517**, 200-204, doi:10.1038/nature13871 (2015).
- 6 Ramanathan, K. R., Jin, J., Giustino, T. F., Payne, M. R. & Maren, S. Prefrontal projections to the thalamic nucleus reuniens mediate fear extinction. *Nat Commun* **9**, 4527, doi:10.1038/s41467-018-06970-z (2018).
- 7 Ramanathan, K. R. & Maren, S. Nucleus reuniens mediates the extinction of contextual fear conditioning. *Behav Brain Res* **374**, 112114, doi:10.1016/j.bbr.2019.112114 (2019).

REVIEWERS' COMMENTS

Reviewer #1 (Remarks to the Author):

The authors have satisfactorily addressed the concerns expressed in the previous round of review; the paper represents a timely and important contribution to the field.

Reviewer #2 (Remarks to the Author):

The authors have responded well to my previous comments including my concerns regarding the interpretations of the NR-CA1 pathway in memory retrieval and suppression. I also believe the authors' responses to the other reviewer comments were adequate. I have no further concerns. The results are noteworthy for the field especially the NR-axonal imaging data, which speak directly to the physiological dynamics at play in the critical NR-CA1 projections during memory retrieval.

Reviewer #3 (Remarks to the Author):

I am satisfied by the response of the authors to the reviewer comments from the previous review at Nature Neuroscience. I feel these results are highly noteworthy and significant for demonstrating the role of Nucleus Reuniens axon projections to region CA1 in suppressing freezing associated with contextual fear conditioning. The data effectively supports the conclusions and the authors have effectively responded to all the reviewer comments from the previous round of reviews.

Reviewer #4 (Remarks to the Author):

The authors have significantly modified and much improved their original manuscript, with quite extensive changes in the text, the addition of figures and in depth responses to the comments of the four reviewers.

I am essentially satisfied with their response to my comments, with a few considerations:

1. The DREADD injection in RE (of Fig. 2B) is very unclear, the section is not visible

2. They acknowledge the difficulty in tracking the same axon across days and in a clarification of this point indicate they were only able to do this in 4 of 10 mice. While this probably suffices, it is important to clearly state this (4 of 10 mice) in the Results/Discussion.

3. I probably still question the seemingly opposing findings of their DREADD and axon experiments but then again they fully explain their position.

REVIEWERS' COMMENTS

Reviewer #1 (Remarks to the Author):

The authors have satisfactorily addressed the concerns expressed in the previous round of review; the paper represents a timely and important contribution to the field.

We thank the reviewer for their contributions to this manuscript.

Reviewer #2 (Remarks to the Author):

The authors have responded well to my previous comments including my concerns regarding the interpretations of the NR-CA1 pathway in memory retrieval and suppression. I also believe the authors' responses to the other reviewer comments were adequate. I have no further concerns. The results are noteworthy for the field especially the NR-axonal imaging data, which speak directly to the physiological dynamics at play in the critical NR-CA1 projections during memory retrieval.

We thank the reviewer for their contributions to this manuscript.

Reviewer #3 (Remarks to the Author):

I am satisfied by the response of the authors to the reviewer comments from the previous review at Nature Neuroscience. I feel these results are highly noteworthy and significant for demonstrating the role of Nucleus Reuniens axon projections to region CA1 in suppressing freezing associated with contextual fear conditioning. The data effectively supports the conclusions and the authors have effectively responded to all the reviewer comments from the previous round of reviews.

We thank the reviewer for their contributions to this manuscript.

Reviewer #4 (Remarks to the Author):

The authors have significantly modified and much improved their original manuscript, with quite extensive changes in the text, the addition of figures and in depth responses to the comments of the four reviewers.

I am essentially satisfied with their response to my comments, with a few considerations:

1. The DREADD injection in RE (of Fig. 2B) is very unclear, the section is not visible
2. They acknowledge the difficulty in tracking the same axon across days and in a clarification of this point indicate they were only able to do this in 4 of 10 mice. While this probably suffices, it is important to clearly state this (4 of 10 mice) in the Results/Discussion.

3. I probably still question the seemingly opposing findings of their DREADD and axon experiments but then again they fully explain their position.

1. With due considerations of the reviewer's comments, in light of manuscript figure size requirements and other reviewer comments, we believe the DREADD injection is visible in this figure and adequately conveys an example location of injection. We thank the reviewer for their comment.

2. This is clearly stated in the Results section, in the figure legend for figure 5, and in the Methods. We thank the reviewer for their comment.

3. As the reviewer noted, this concern has already been addressed in the previous review round. We thank the reviewer for their comment.